# Generative Modeling via Drifting

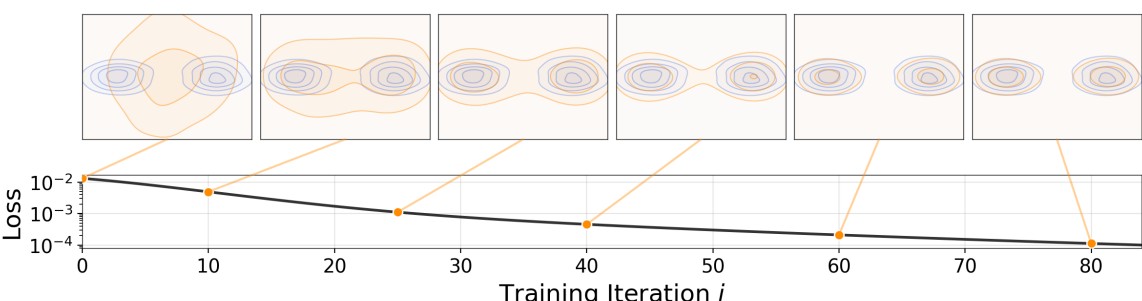

*Figure 1.* **Drifting Model.** A network $f$ performs a pushforward operation: $q = f_\# p_{\text{prior}}$, mapping a prior distribution $p_{\text{prior}}$ (*e.g.*, Gaussian, not shown here) to a pushforward distribution $q$ (orange). The goal of training is to approximate the data distribution $p_{\text{data}}$ (blue). As training iterates, we obtain a sequence of models $\{f_i\}$, which corresponds to a sequence of pushforward distributions $\{q_i\}$. Our Drifting Model focuses on the evolution of this pushforward distribution at *training-time*. We introduce a drifting field (detailed in main text) that approaches zero when $q$ matches $p_{\text{data}}$. This drifting field provides a loss function (y-axis, in log-scale) for training.

## Abstract

Generative modeling can be formulated as learning a mapping $f$ such that its pushforward distribution matches the data distribution. The pushforward behavior can be carried out iteratively at inference time, *e.g.*, in diffusion/flow-based models. In this paper, we propose a new paradigm called *Drifting Models*, which evolve the pushforward distribution during training and naturally admit one-step inference. We introduce a drifting field that governs the sample movement and achieves equilibrium when the distributions match. This leads to a training objective that allows the neural network optimizer to evolve the distribution. In experiments, our one-step generator achieves state-of-the-art results on ImageNet 256×256, with FID 1.54 in latent space and 1.73 in pixel space. We hope that our work opens up new opportunities for high-quality one-step generation.

## 1. Introduction

Generative models are commonly regarded as more challenging than discriminative models. While discriminative modeling typically focuses on mapping individual samples to their corresponding labels, generative modeling concerns mapping from one distribution to another. This can be expressed as learning a mapping $f$ such that the *pushforward* of a prior distribution $p_{\text{prior}}$ matches the data distribution, namely, $f_\# p_{\text{prior}} \approx p_{\text{data}}$. Conceptually, generative modeling learns a *functional* (here, $f_\#$) that maps from one function (here, a distribution) to another.

The "pushforward" behavior can be realized *iteratively* at *inference* time, *e.g.*, in prevailing paradigms such as Diffusion (Sohl-Dickstein et al., 2015) and Flow Matching (Lipman et al., 2022). When generating, these models map noisier samples to slightly cleaner ones, progressively evolving the sample distribution toward the data distribution. This modeling philosophy can be viewed as decomposing a complex pushforward map (*i.e.*, $f_\#$) into a chain of more feasible transformations, applied at inference time.

In this paper, we propose *Drifting Models*, a new paradigm for generative modeling. Drifting Models are characterized by learning a pushforward map that evolves during *training* time, thereby removing the need for an iterative inference procedure. The mapping $f$ is represented by a single-pass, non-iterative network. As the training process is inherently iterative in deep learning optimization, it can be naturally viewed as evolving the pushforward distribution, $f_\# p_{\text{prior}}$, through the update of $f$. See Fig. 1.

To drive the evolution of the training-time pushforward, we introduce a *drifting field* that governs the sample movement. This field depends on the generated distribution and the data distribution. By definition, this field becomes zero when the two distributions match, thereby reaching an equilibrium in

[1]Anonymous Institution, Anonymous City, Anonymous Region, Anonymous Country. Correspondence to: Anonymous Author <anon.email@domain.com>.

Preliminary work. Under review by the International Conference on Machine Learning (ICML). Do not distribute.

which the samples no longer drift.

Building on this formulation, we propose a simple training objective that minimizes the *drift* of the generated samples. This objective induces sample movements and thereby evolves the underlying pushforward distribution through iterative optimization (*e.g.*, SGD). We further introduce the designs of the drifting field, the neural network model, and the training algorithm.

Drifting Models naturally perform *single-step* ("1-NFE") generation and achieve strong empirical performance. On ImageNet 256×256, we obtain a 1-NFE FID of **1.54** under the standard latent-space generation protocol, achieving a new state-of-the-art among single-step methods. This result remains competitive even when compared with *multi-step* diffusion-/flow-based models. Further, under the more challenging *pixel*-space generation protocol (*i.e.*, without latents), we reach a 1-NFE FID of **1.73**, substantially outperforming previous pixel-space methods. These results suggest that Drifting Models offer a promising new paradigm for high-quality, efficient generative modeling.

## 2. Related Work

**Diffusion-/Flow-based Models.** Diffusion models (*e.g.*, Sohl-Dickstein et al. 2015; Ho et al. 2020; Song et al. 2020) and their flow-based counterparts (*e.g.*, Lipman et al. 2022; Liu et al. 2022; Albergo et al. 2023) formulate noise-to-data mappings through *differential equations* (SDEs or ODEs). At the core of their inference-time computation is an *iterative* update, *e.g.*, of the form $\mathbf{x}_{i+1} = \mathbf{x}_i + \Delta\mathbf{x}_i$, such as with an Euler solver. The update $\Delta\mathbf{x}_i$ depends on the neural network $f$, and as a result, generation involves multiple steps of network evaluations.

A growing body of work has focused on reducing the steps of diffusion-/flow-based models. Distillation-based methods (*e.g.*, Salimans & Ho 2022; Luo et al. 2023; Yin et al. 2024; Zhou et al. 2024) distill a pretrained multi-step model into a single-step one. Another line of research aims to train one-step diffusion/flow models from scratch (*e.g.*, Song et al. 2023; Frans et al. 2024; Boffi et al. 2025; Geng et al. 2025a). To achieve this goal, these methods incorporate the SDE/ODE dynamics into training by approximating the induced trajectories. In contrast, our work presents a conceptually different paradigm and does not rely on SDE/ODE formulations as in diffusion/flow models.

**Generative Adversarial Networks (GANs).** GANs (Goodfellow et al., 2014) are a classical family of models that train a generator by discriminating generated samples from real data. Like GANs, our method involves a single-pass network $f$ that maps noise to data, whose "goodness" is evaluated by a loss function; however, unlike GANs, our method does not rely on adversarial optimization.

**Variational Autoencoders (VAEs).** VAEs (Kingma & Welling, 2013) optimize the evidence lower bound (ELBO), which consists of a reconstruction loss and a KL divergence term. Classical VAEs are one-step generators when using a Gaussian prior. Today's prevailing VAE applications often resort to priors learned from other methods, *e.g.*, diffusion (Rombach et al., 2022) or autoregressive models (Esser et al., 2021), where VAEs effectively act as tokenizers.

**Normalizing Flows (NFs).** NFs (Rezende & Mohamed, 2015; Dinh et al., 2016; Zhai et al., 2024) learn mappings from data to noise and optimize the log-likelihood of samples. These methods require invertible architectures and computable Jacobians. Conceptually, NFs operate as one-step generators at inference, with computation performed by the inverse of the network.

**Moment Matching.** Moment-matching methods (Dziugaite et al., 2015; Li et al., 2015) seek to minimize the Maximum Mean Discrepancy (MMD) between the generated and data distributions. Moment Matching has recently been extended to one-/few-step diffusion (Zhou et al., 2025). Related to MMD, our approach also leverages the concepts of kernel functions and positive/negative samples. However, our approach focuses on a drifting field that explicitly governs the sample drifts at training time.

**Contrastive Learning.** Our drifting field is driven by positive samples from the data distribution and negative samples from the generated distribution. This is conceptually related to the positive and negative samples in *contrastive representation learning* (Hadsell et al., 2006; Oord et al., 2018). The idea of contrastive learning has also been extended to generative models, *e.g.*, to GANs (Unterthiner et al., 2017; Kang & Park, 2020) or Flow Matching (Stoica et al., 2025).

## 3. Drifting Models for Generation

We propose Drifting Models, which formulate generative modeling as a *training-time* evolution of the pushforward distribution via a drifting field. Our model naturally performs one-step generation at inference time.

### 3.1. Pushforward at Training Time

Consider a neural network $f : \mathbb{R}^C \mapsto \mathbb{R}^D$. The input of $f$ is $\boldsymbol{\epsilon} \sim p_{\boldsymbol{\epsilon}}$ (*e.g.*, any noise of dimension $C$), and the output is denoted by $\mathbf{x} = f(\boldsymbol{\epsilon}) \in \mathbb{R}^D$. In general, the input and output dimensions need not be equal.

We denote the distribution of the network output by $q$, *i.e.*, $\mathbf{x} = f(\boldsymbol{\epsilon}) \sim q$. In probability theory, $q$ is referred to as the *pushforward* distribution of $p_{\boldsymbol{\epsilon}}$ under $f$, denoted by:

$$q = f_{\#}p_{\boldsymbol{\epsilon}}. \tag{1}$$

Here, "$f_{\#}$" denotes the pushforward induced by $f$. Intu-

itively, this notation means that $f$ transforms a distribution $p_\epsilon$ into another distribution $q$. The goal of generative modeling is to find $f$ such that $f_\# p_\epsilon \approx p_{\text{data}}$.

Since neural network *training* is inherently iterative (*e.g.*, SGD), the training process produces a sequence of models $\{f_i\}$, where $i$ denotes the training iteration. This corresponds to a sequence of pushforward distributions $\{q_i\}$ during training, where $q_i = [f_i]_\# p_\epsilon$ for each $i$. The training process progressively evolves $q_i$ to match $p_{\text{data}}$.

When the network $f$ is updated, a sample at training iteration $i$ is implicitly "drifted" as: $\mathbf{x}_{i+1} = \mathbf{x}_i + \Delta \mathbf{x}_i$, where $\Delta \mathbf{x}_i := f_{i+1}(\epsilon) - f_i(\epsilon)$ arises from parameter updates to $f$. This implies that the update of $f$ determines the "*residual*" of $\mathbf{x}$, which we refer to as the "drift".

### 3.2. Drifting Field for Training

Next, we define a ***drifting field*** to govern the training-time evolution of the samples $\mathbf{x}$ and, consequently, the pushforward distribution $q$. A drifting field is a function that computes $\Delta \mathbf{x}$ given $\mathbf{x}$. Formally, denoting this field by $\mathbf{V}_{p,q}(\cdot)\colon \mathbb{R}^d \to \mathbb{R}^d$, we have:

$$\mathbf{x}_{i+1} = \mathbf{x}_i + \mathbf{V}_{p,q_i}(\mathbf{x}_i), \tag{2}$$

Here, $\mathbf{x}_i = f_i(\epsilon) \sim q_i$ and after drifting we denote $\mathbf{x}_{i+1} \sim q_{i+1}$. The subscripts $p, q$ denote that this field depends on $p$ (*e.g.*, $p = p_{\text{data}}$) and the current distribution $q$.

Ideally, when $p = q$, we want all $\mathbf{x}$ to stop drifting *i.e.*, $\mathbf{V} = \mathbf{0}$. In this paper, we consider the following proposition:

**Proposition 3.1.** *Consider an **anti-symmetric** drifting field:*

$$\mathbf{V}_{p,q}(\mathbf{x}) = -\mathbf{V}_{q,p}(\mathbf{x}), \quad \forall \mathbf{x}. \tag{3}$$

*Then we have:* $\quad q = p \quad \Rightarrow \quad \mathbf{V}_{p,q}(\mathbf{x}) = \mathbf{0}, \forall \mathbf{x}.$

The proof is straightforward[1]. Intuitively, anti-symmetry means that swapping $p$ and $q$ simply flips the sign of the drift. This proposition implies that if the pushforward distribution $q$ matches the data distribution $p$, the drift is zero for any sample and the model achieves an equilibrium.

We note that the converse implication, *i.e.*, $\mathbf{V}_{p,q} = \mathbf{0} \Rightarrow q = p$, is false in general for arbitrary choices of $\mathbf{V}$. For our kernelized formulation (Sec. 3.3), we give sufficient conditions under which $\mathbf{V}_{p,q} \approx \mathbf{0}$ implies $q \approx p$ (Appendix C).

**Training Objective.** The property of equilibrium motivates a definition of a training objective. Let $f_\theta$ be a network parameterized by $\theta$, and $\mathbf{x} = f_\theta(\epsilon)$ for $\epsilon \sim p_\epsilon$. At the equilibrium where $\mathbf{V} = \mathbf{0}$, we set up the following *fixed-point* relation:

$$f_{\hat\theta}(\epsilon) = f_{\hat\theta}(\epsilon) + \mathbf{V}_{p,q_{\hat\theta}}\big(f_{\hat\theta}(\epsilon)\big). \tag{4}$$

---

[1] $q = p \Rightarrow \mathbf{V}_{p,q} = \mathbf{V}_{q,p} = -\mathbf{V}_{p,q} \Rightarrow \mathbf{V}_{p,q} = \mathbf{0}$

Here, $\hat\theta$ denotes the optimal parameters that can achieve the equilibrium, and $q_{\hat\theta}$ denotes the pushforward of $f_{\hat\theta}$.

This equation motivates a fixed-point iteration during training. At iteration $i$, we seek to satisfy:

$$f_{\theta_{i+1}}(\epsilon) \leftarrow f_{\theta_i}(\epsilon) + \mathbf{V}_{p,q_{\theta_i}}\big(f_{\theta_i}(\epsilon)\big). \tag{5}$$

We convert this update rule into a loss function:

$$\mathcal{L} = \mathbb{E}_\epsilon \Big[ \big\| \underbrace{f_\theta(\epsilon)}_{\text{prediction}} - \underbrace{\texttt{stopgrad}\big(f_\theta(\epsilon) + \mathbf{V}_{p,q_\theta}\big(f_\theta(\epsilon)\big)\big)}_{\text{frozen target}} \big\|^2 \Big]. \tag{6}$$

Here, the stop-gradient operation provides a frozen state from the last iteration, following (Chen & He, 2021; Song & Dhariwal, 2023). Intuitively, we compute a frozen target and move the network prediction toward it.

We note that the *value* of our loss function $\mathcal{L}$ is equal to $\mathbb{E}_\epsilon\big[\|\mathbf{V}(f(\epsilon))\|^2\big]$, that is, the squared norm of the drifting field $\mathbf{V}$. With the stop-gradient formulation, our solver does not directly back-propagate through $\mathbf{V}$, because $\mathbf{V}$ depends on $q_\theta$ and back-propagating through a distribution is non-trivial. Instead, our formulation minimizes this objective *indirectly*: it moves $\mathbf{x} = f_\theta(\epsilon)$ towards its drifted version, *i.e.*, towards $\mathbf{x} + \Delta \mathbf{x}$ that is frozen at this iteration.

### 3.3. Designing the Drifting Field

The field $\mathbf{V}_{p,q}$ depends on two distributions $p$ and $q$. To obtain a computable formulation, we consider the form:

$$\mathbf{V}_{p,q}(\mathbf{x}) = \mathbb{E}_{\mathbf{y}^+ \sim p} \mathbb{E}_{\mathbf{y}^- \sim q}[\mathcal{K}(x, \mathbf{y}^+, \mathbf{y}^-)], \tag{7}$$

where $\mathcal{K}(\cdot, \cdot, \cdot)$ is a kernel-like function describing interactions among three sample points. $\mathcal{K}$ can optionally depend on $p$ and $q$. Our framework supports a broad class of functions $\mathcal{K}$, as long as $\mathbf{V} = 0$ when $p = q$.

For the instantiation in this work, we introduce a form of $\mathbf{V}$ driven by attraction and repulsion. We define the following fields inspired by the *mean-shift* method (Cheng, 1995):

$$\begin{aligned} \mathbf{V}_p^+(\mathbf{x}) &:= \frac{1}{Z_p} \mathbb{E}_p\big[k(\mathbf{x}, \mathbf{y}^+)(\mathbf{y}^+ - \mathbf{x})\big], \\ \mathbf{V}_q^-(\mathbf{x}) &:= \frac{1}{Z_q} \mathbb{E}_q\big[k(\mathbf{x}, \mathbf{y}^-)(\mathbf{y}^- - \mathbf{x})\big]. \end{aligned} \tag{8}$$

Here, $Z_p$ and $Z_q$ are normalization factors:

$$\begin{aligned} Z_p(\mathbf{x}) &:= \mathbb{E}_p[k(\mathbf{x}, \mathbf{y}^+)], \\ Z_q(\mathbf{x}) &:= \mathbb{E}_q[k(\mathbf{x}, \mathbf{y}^-)]. \end{aligned} \tag{9}$$

Intuitively, Eq. (8) computes the weighted mean of the vector difference $\mathbf{y} - \mathbf{x}$. The weights are given by a kernel $k(\cdot, \cdot)$ normalized by (9). We then define $\mathbf{V}$ as:

$$\mathbf{V}_{p,q}(\mathbf{x}) := \mathbf{V}_p^+(\mathbf{x}) - \mathbf{V}_q^-(\mathbf{x}). \tag{10}$$

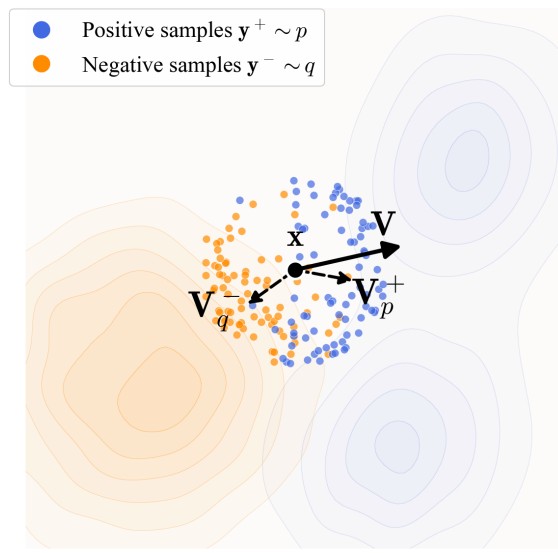

*Figure 2.* **Illustration of drifting a sample.** A generated sample $\mathbf{x}$ (black) drifts according to a vector: $\mathbf{V} = \mathbf{V}_p^+ - \mathbf{V}_q^-$. Here, $\mathbf{V}_p^+$ is the mean-shift vector of the positive samples (blue) and $\mathbf{V}_q^-$ is the mean-shift vector of the negative samples (orange): see Eq. (8). $\mathbf{x}$ is attracted by $\mathbf{V}_p^+$ and repulsed by $\mathbf{V}_q^-$.

Intuitively, this field can be viewed as attracting by the data distribution $p$ and repulsing by the sample distribution $q$. This is illustrated in Fig. 2.

Substituting Eq. (8) into Eq. (10), we obtain:

$$\mathbf{V}_{p,q}(\mathbf{x}) = \frac{1}{Z_p Z_q} \mathbb{E}_{p,q}\big[k(\mathbf{x}, \mathbf{y}^+)k(\mathbf{x}, \mathbf{y}^-)(\mathbf{y}^+ - \mathbf{y}^-)\big]. \quad (11)$$

Here, the vector difference reduces to $\mathbf{y}^+ - \mathbf{y}^-$; the weight is computed from two kernels and normalized jointly. This form is an instantiation of Eq. (7). It is easy to see that $\mathbf{V}$ is anti-symmetric: $\mathbf{V}_{p,q} = -\mathbf{V}_{q,p}$. In general, our method does not require $\mathbf{V}$ to be decomposed into attraction and repulsion; it only requires $\mathbf{V} = 0$ when $p = q$.

**Kernel.** The kernel $k(\cdot, \cdot)$ can be a function that measures the similarity. In this paper, we adopt:

$$k(\mathbf{x}, \mathbf{y}) = \exp\left(-\frac{1}{\tau}\|\mathbf{x} - \mathbf{y}\|\right), \quad (12)$$

where $\tau$ is a temperature and $\|\cdot\|$ is $\ell_2$-distance. We view $\tilde{k}(\mathbf{x}, \mathbf{y}) \triangleq \frac{1}{Z}k(\mathbf{x}, \mathbf{y})$ as a normalized kernel, which absorbs the normalization in Eq. (11).

In practice, we implement $\tilde{k}$ using a *softmax* operation, with logits given by $-\frac{1}{\tau}\|\mathbf{x} - \mathbf{y}\|$, where the softmax is taken over $\mathbf{y}$. This softmax operation is similar to that of InfoNCE (Oord et al., 2018) in contrastive learning. In our implementation, we further apply an extra softmax normalization over the set of $\{\mathbf{x}\}$ within a batch, which slightly improves performance in practice. This additional normalization does not alter the antisymmetric property of the resulting $\mathbf{V}$.

**Algorithm 1 Training Loss.** Note: for brevity, here the negative samples y_neg are from the same batch of generated data, though they can include other source of negatives.

```
# f: generator
# y_pos: [N_pos, D], data samples

e = randn([N, C]) # noise
x = f(e) # [N, D], generated samples
y_neg = x # reuse x as negatives

V = compute_V(x, y_pos, y_neg)
x_drifted = stopgrad(x + V)

loss = mse_loss(x - x_drifted)
```

**Equilibrium and Matched Distributions.** Since our training loss in Eq. (6) encourages minimizing $\|\mathbf{V}\|^2$, we hope that $\mathbf{V} \approx \mathbf{0}$ leads to $q \approx p$. While this implication does not hold for arbitrary choices of $\mathbf{V}$, we empirically observe that decreasing the value of $\|\mathbf{V}\|^2$ correlates with improved generation quality. In Appendix C, we provide an identifiability heuristic: for our kernelized construction, the zero-drift condition imposes a large set of bilinear constraints on $(p, q)$, and under mild non-degeneracy assumptions this forces $p$ and $q$ to match (approximately).

**Stochastic Training.** In stochastic training (*e.g.*, mini-batch optimization), we estimate $\mathbf{V}$ by approximating the expectations in Eq. (11) with empirical means. For each training step, we draw $N$ samples of noise $\boldsymbol{\epsilon} \sim p_\epsilon$ and compute a batch of $\mathbf{x} = f_\theta(\boldsymbol{\epsilon}) \sim q$. The generated samples also serve as the negative samples in the same batch, *i.e.*, $\mathbf{y}^- \sim q$. On the other hand, we sample $N_{\text{pos}}$ data points $\mathbf{y}^+ \sim p_{\text{data}}$. The drifting field $\mathbf{V}$ is computed in this batch of positive and negative samples. Alg. 1 provide the pseudocode for such a training step, where compute_V is given in Appendix A.1.

### 3.4. Drifting in Feature Space

Thus far, we have defined the objective (6) directly in the raw data space. Our formulation can be extended to any feature space. Let $\phi$ denote a feature extractor (*e.g.*, an image encoder) operating on real or generated samples. We rewrite the loss (6) in the feature space as:

$$\mathbb{E}\left[\left\|\phi(\mathbf{x}) - \texttt{stopgrad}\big(\phi(\mathbf{x}) + \mathbf{V}\big(\phi(\mathbf{x})\big)\big)\right\|^2\right]. \quad (13)$$

Here, $\mathbf{x} = f_\theta(\boldsymbol{\epsilon})$ is the output (*e.g.*, images) of the generator. $\mathbf{V}$ is defined in the feature space: in practice, this means that $\phi(\mathbf{y}^+)$ and $\phi(\mathbf{y}^-)$ serve as the positive/negative samples. It is worth noting that feature encoding is a training-time operation and is not used at inference time.

This can be further extended to multiple features, *e.g.*, at

multiple scales and locations:

$$\sum_j \mathbb{E}\left[\left\|\phi_j(\mathbf{x}) - \texttt{stopgrad}\Big(\phi_j(\mathbf{x}) + \mathbf{V}\big(\phi_j(\mathbf{x})\big)\Big)\right\|_2^2\right]. \tag{14}$$

Here, $\phi_j$ represents the feature vectors at the $j$-th scale and/or location from an encoder $\phi$. With a ResNet-style image encoder (He et al., 2016), we compute drifting losses across multiple scales and locations, which provides richer gradient information for training.

The feature extractor plays an important role in the generation of high-dimensional data. As our method is based on the kernel $k(\cdot,\cdot)$ for characterizing sample similarities, it is desired for semantically similar samples to stay close in the feature space. This goal is aligned with self-supervised learning (*e.g.*, He et al. 2020; Chen et al. 2020a). We use pre-trained self-supervised models as the feature extractor.

**Relation to Perceptual Loss.** Our feature-space loss is related to perceptual loss (Zhang et al., 2018) but is conceptually different. The perceptual loss minimizes: $\|\phi(\mathbf{x}) - \phi(\mathbf{x}_{\text{target}})\|_2^2$, that is, the regression target is $\phi(\mathbf{x}_{\text{target}})$ and requires pairing $\mathbf{x}$ with its target. In contrast, our regression target in (13) is $\phi(\mathbf{x}) + \mathbf{V}\big(\phi(\mathbf{x})\big)$, where the drifting is in the feature space and requires no pairing. In principle, our feature-space loss aims to match the pushforward distributions $\phi_\# q$ and $\phi_\# p$.

**Relation to Latent Generation.** Our feature-space loss is *orthogonal* to the concept of generators in the latent space (*e.g.*, Latent Diffusion (Rombach et al., 2022)). In our case, when using $\phi$, the generator $f$ can still produce outputs in the pixel space or the latent space of a tokenizer. If the generator $f$ is in the latent space and the feature extractor $\phi$ is in the pixel space, the tokenizer decoder is applied before extracting features from $\phi$.

### 3.5. Classifier-Free Guidance

Classifier-free guidance (CFG) (Ho & Salimans, 2022) improves generation quality by extrapolating between class-conditional and unconditional distributions. Our method naturally supports a related form of guidance.

In our model, given a class label $c$ as the condition, the underlying target distribution $p$ now becomes $p_{\text{data}}(\cdot|c)$, from which we can draw positive samples: $\mathbf{y}^+ \sim p_{\text{data}}(\cdot|c)$. To achieve guidance, we draw negative samples either from generated samples or *real* samples from different classes. Formally, the negative sample distribution is now:

$$\tilde{q}(\cdot|c) \triangleq (1-\gamma)\, q_\theta(\cdot|c) + \gamma\, p_{\text{data}}(\cdot|\varnothing). \tag{15}$$

Here, $\gamma \in [0,1)$ is a mixing rate, and $p_{\text{data}}(\cdot|\varnothing)$ denotes the *unconditional* data distribution[2].

The goal of learning is to find $\tilde{q}(\cdot|c) = p_{\text{data}}(\cdot|c)$. Substitut-

ing it into (15), we obtain:

$$q_\theta(\cdot|c) = \alpha\, p_{\text{data}}(\cdot|c) - (\alpha-1)\, p_{\text{data}}(\cdot|\varnothing). \tag{16}$$

where $\alpha = \frac{1}{1-\gamma} \geq 1$. This implies that $q_\theta(\cdot|c)$ is to approximate a linear combination of conditional and unconditional data distributions. This follows the spirit of original CFG.

In practice, Eq. (15) means that we sample extra negative examples from the data in $p_{\text{data}}(\cdot|\varnothing)$, in addition to the generated data. The distribution $q_\theta(\cdot|c)$ corresponds to a class-conditional network $f_\theta(\cdot|c)$, similar to common practice (Ho & Salimans, 2022). We note that, in our method, CFG is a *training-time* behavior by design: the one-step (1-NFE) property is preserved at inference time.

## 4. Implementation for Image Generation

We describe our implementation for image generation on ImageNet (Deng et al., 2009) at resolution 256×256. Full implementation details are provided in Appendix A.

**Tokenizer.** By default, we perform generation in latent space (Rombach et al., 2022). We adopt the standard SD-VAE tokenizer, which produces a 32×32×4 latent space in which generation is performed.

**Architecture.** Our generator ($f_\theta$) has a DiT-like (Peebles & Xie, 2023) architecture. Its input is 32×32×4-dim Gaussian noise $\epsilon$, and its output is the generated latent $\mathbf{x}$ of the same dimension. We use a patch size of 2, *i.e.*, like DiT/2. Our model uses adaLN-zero (Peebles & Xie, 2023) for processing class-conditioning or other extra conditioning.

**CFG conditioning.** We follow (Geng et al., 2025b) and adopt CFG-conditioning. At training time, a CFG scale $\alpha$ (Eq. (16)) is randomly sampled. Negative samples are prepared based on $\alpha$ (Eq. (15)), and the network is conditioned on this value. At inference time, $\alpha$ can be freely specified and varied without retraining. Details are in A.7.

**Batching.** The pseudo-code in Alg. 1 describes a batch of $N = N_{\text{neg}}$ generated samples. In practice, when class labels are involved, we sample a batch of $N_{\text{c}}$ class labels. For each label, we perform Alg. 1 *independently*. Accordingly, the *effective* batch size is $B = N_{\text{c}} \times N$, which consists of $N_{\text{c}} \times N$ negatives and $N_{\text{c}} \times N_{\text{pos}}$ positives.

We define a "training epoch" based on the number of generated samples $\mathbf{x}$. In particular, each iteration generates $B$ samples, and one epoch corresponds to $N_{\text{data}}/B$ iterations for a dataset of size $N_{\text{data}}$.

**Feature Extractor.** Our model is trained with drifting loss in a feature space (Sec. 3.4). The feature extractor $\phi$ is an image encoder. We mainly consider a ResNet-style (He

---

[2]This should be the data distribution excluding the class $c$. For simplicity, we use the unconditional data distribution.

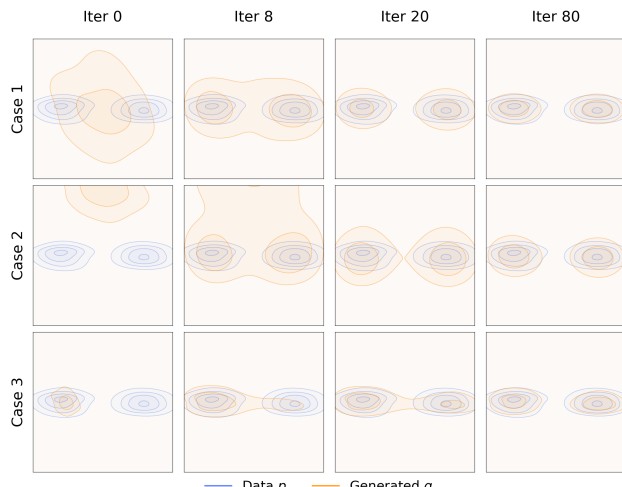

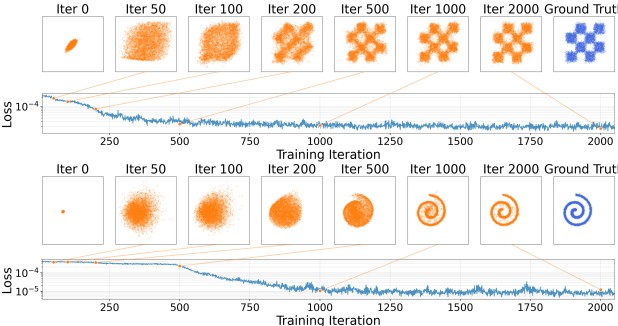

*Figure 4.* **Evolution of samples.** We show generated points sampled at different training iterations, along with their loss values. The loss (whose value equals $\|V\|^2$) decreases as the distribution converges to the target. (y-axis is log-scale.)

*Figure 3.* **Evolution of the generated distribution.** The distribution $q$ (orange) evolves toward a bimodal target $p$ (blue) during training. We show three initializations of $q$: **(top)**: initialized between the two modes; **(middle)**: initialized far from both modes; **(bottom)**: initialized collapsed onto one mode. Across all initializations, our method approximates the target distribution without mode collapse.

et al., 2016) encoder, pre-trained by self-supervised learning, *e.g.*, MoCo (He et al., 2020) and SimCLR (Chen et al., 2020a). When these pre-trained models operate in pixel space, we apply the VAE decoder to map our generator's latent-space output back to pixel space for feature extraction. Gradients are backpropagated through the feature encoder and VAE decoder. We also study an MAE (He et al., 2022) pre-trained in latent space (detailed in A.3).

For all ResNet-style models, features are extracted from multiple stages (*i.e.*, multi-scale feature maps). The drifting loss in (13) is computed at each scale and then combined. We elaborate on the details in A.6.

**Pixel-space Generation.** While our experiments primarily focus on latent-space generation, our models support pixel-space generation. In this case, $\epsilon$ and $\mathbf{x}$ are both $256 \times 256 \times 3$. We use a patch size of 16 (*i.e.*, DiT/16). The feature extractor $\phi$ is directly on the pixel space.

# 5. Experiments

## 5.1. Toy Experiments

**Evolution of the generated distribution.** Figure 3 visualizes a 2D toy case, where $q$ evolves toward a bimodal distribution $p$ at training time, under three initializations.

In this toy example, our method approximates the target distribution without exhibiting mode collapse. This holds even when $q$ is initialized in a collapsed single-mode state (bottom). This provides intuition into why our method is robust to mode collapse: if $q$ collapses onto one mode,

*Table 1.* **Importance of anti-symmetry:** breaking the anti-symmetry leads to failure. Here, the anti-symmetric case is defined in Eq. (10) and Eq. (11); other destructive cases are defined in similar ways. (Setting: B/2 model, 100 epochs)

| case | drifting field $\mathbf{V}$ | FID |
|---|---|---|
| **anti-symmetry** (default) | $\mathbf{V}^+ - \mathbf{V}^-$ | **8.46** |
| $1.5\times$ attraction | $1.5\mathbf{V}^+ - \mathbf{V}^-$ | 41.05 |
| $1.5\times$ repulsion | $\mathbf{V}^+ - 1.5\mathbf{V}^-$ | 46.28 |
| $2.0\times$ attraction | $2\mathbf{V}^+ - \mathbf{V}^-$ | 86.16 |
| $2.0\times$ repulsion | $\mathbf{V}^+ - 2\mathbf{V}^-$ | 112.84 |
| attraction-only | $\mathbf{V}^+$ | 177.14 |

other modes of $p$ will attract the samples, allowing them to continue moving and pushing $q$ to continue evolving.

**Evolution of the samples.** Figure 4 shows the training process on two 2D cases. A small MLP generator is trained. The loss (whose value equals $\|\mathbf{V}\|^2$) decreases as the generated distribution converges to the target. This is in line with our motivation that reducing the drift and pushing towards the equilibrium will approximately yield $p = q$.

## 5.2. ImageNet Experiments

We evaluate our models on ImageNet $256 \times 256$. Ablation studies use a B/2 model on the SD-VAE latent space, trained for 100 epochs. The drifting loss is in a feature space computed by a latent-MAE encoder. We report FID (Heusel et al., 2017) on 50K generated images. We analyze the results as follows.

**Anti-symmetry.** Our derivation of equilibrium requires the drifting field to be anti-symmetric; see Eq. (3). In Table 1, we conduct a *destructive* study that intentionally breaks this anti-symmetry. The anti-symmetric case (our ablation default) works well, while other cases fail catastrophically.

Intuitively, for a sample $\mathbf{x}$, we want attraction from $p$ to be canceled by repulsion from $q$ when $p$ and $q$ match. This equilibrium is not achieved in the destructive cases.

*Table 2.* **Allocation of positive and negative samples.** In both sub-tables, we control the total compute by fixing the epochs (100) and the batch size $B = N_c \times N_{pos}$ (4096). Here, $N_c$ is for class labels. Under the same budget, increasing positive samples (**left**) and negative samples (**right**) improves generation quality. (Setting: B/2 model, 100 epochs)

| $N_c$ | $N_{pos}$ | $N_{neg}$ | $B$ | FID | | $N_c$ | $N_{pos}$ | $N_{neg}$ | $B$ | FID |
|---|---|---|---|---|---|---|---|---|---|---|
| 64 | 1 | 64 | 4096 | 20.43 | | 512 | 8 | 8 | 4096 | 11.82 |
| 64 | 16 | 64 | 4096 | 10.39 | | 256 | 16 | 16 | 4096 | 10.16 |
| 64 | 32 | 64 | 4096 | 8.97 | | 128 | 32 | 32 | 4096 | 9.32 |
| 64 | **64** | 64 | 4096 | **8.46** | | 64 | 64 | **64** | 4096 | **8.46** |

*Table 3.* **Feature space for drifting.** We compare self-supervised learning (SSL) encoders. Standard SimCLR and MoCo encoders achieve competitive results, whereas our customized latent-MAE performs best and benefits from increased width and longer training. (Generator setting: B/2 model, 100 epochs)

| | | feature encoder ($\phi$) | | | | |
|---|---|---|---|---|---|---|
| SSL method | arch | block | width | SSL ep. | FID |
|---|---|---|---|---|---|
| SimCLR | ResNet | bottleneck | 256 | 800 | 11.05 |
| MoCo-v2 | ResNet | bottleneck | 256 | 800 | 8.41 |
| latent-MAE (default) | ResNet | basic | 256 | 192 | 8.46 |
| latent-MAE | ResNet | basic | 384 | 192 | 7.26 |
| latent-MAE | ResNet | basic | 512 | 192 | 6.49 |
| latent-MAE | ResNet | basic | 640 | 192 | 6.30 |
| latent-MAE | ResNet | basic | 640 | 1280 | 4.28 |
| latent-MAE + cls ft | ResNet | basic | 640 | 1280 | **3.36** |

*Table 4.* **From ablation to final setting.** We train our model for more epochs, adjust hyper-parameters for this regime, and use a larger model size.

| case | arch | ep | FID |
|---|---|---|---|
| (a) baseline (from Table 3) | B/2 | 100 | 3.36 |
| (b) longer | B/2 | 320 | 2.51 |
| (c) longer + hyper-param. | B/2 | 1280 | 1.75 |
| (d) larger model | L/2 | 1280 | **1.54** |

*Table 5.* **System-level comparison: ImageNet 256×256 generation in latent space.** FID is on 50K images, all reported with CFG if applicable. The parameter numbers are "generator + decoder". All generators are trained from scratch (*i.e.*, not distilled).

| method | space | params | NFE | FID↓ | IS↑ |
|---|---|---|---|---|---|
| *Multi-step Diffusion/Flows* | | | | | |
| DiT-XL/2 (Peebles & Xie, 2023) | SD-VAE | 675M+49M | 250×2 | 2.27 | 278.2 |
| SiT-XL/2 (Ma et al., 2024) | SD-VAE | 675M+49M | 250×2 | 2.06 | 270.3 |
| SiT-XL/2+REPA (Yu et al., 2024) | SD-VAE | 675M+49M | 250×2 | 1.42 | 305.7 |
| LightningDiT-XL/2 (Yao et al., 2025) | VA-VAE | 675M+70M | 250×2 | 1.35 | 295.3 |
| RAE+DiT$^{DH}$-XL/2 (Zheng et al., 2025) | RAE | 839M+415M | 50×2 | **1.13** | 262.6 |
| *Single-step Diffusion/Flows* | | | | | |
| iCT-XL/2 (Song & Dhariwal, 2023) | SD-VAE | 675M | 1 | 34.24 | – |
| Shortcut-XL/2 (Frans et al., 2024) | SD-VAE | 675M | 1 | 10.60 | – |
| MeanFlow-XL/2 (Geng et al., 2025a) | SD-VAE | 676M | 1 | 3.43 | – |
| AdvFlow-XL/2 (Lin et al., 2025) | SD-VAE | 673M | 1 | 2.38 | 284.2 |
| iMeanFlow-XL/2 (Geng et al., 2025b) | SD-VAE | 610M | 1 | 1.72 | 282.0 |
| *Drifting Models* | | | | | |
| **Drifting Model, B/2** | SD-VAE | 133M | 1 | 1.75 | 263.2 |
| **Drifting Model, L/2** | SD-VAE | 463M | 1 | **1.54** | 258.9 |

**Allocation of Positive and Negative Samples.** Our method samples positive and negative examples to estimate $\mathbf{V}$ (see Alg. 1). In Table 2, we study the effect of $N_{pos}$ and $N_{neg}$, under fixed epochs and fixed batch size $B$.

Table 2 shows that using larger $N_{pos}$ and $N_{neg}$ is beneficial. Larger sample sizes are expected to improve the accuracy of the estimated $\mathbf{V}$ and hence the generation quality. This observation aligns with results in contrastive learning (Oord et al., 2018; He et al., 2020; Chen et al., 2020a), in which larger sample sets improve representation learning.

**Feature Space for Drifting.** Our model computes the drifting loss in a feature space (Sec. 3.4). Table 3 compares the feature encoders. Using the public pre-trained encoders from SimCLR (Chen et al., 2020a) and MoCo v2 (Chen et al., 2020b), our method obtains decent results.

These standard encoders operate in the pixel domain, which requires running the VAE decoder at training. To circumvent this, we pre-train a ResNet-style model with the MAE objective (He et al., 2022), directly on the latent space. The feature space produced by this "latent-MAE" performs strongly (Table 3). Increasing the MAE encoder width and the number of pre-training epochs both improve generation quality; fine-tuning it with a classifier ('cls ft') boosts the results further to 3.36 FID.

The comparison in Table 3 shows that the quality of the feature encoder plays an important role. We hypothesize that this is because our method depends on a kernel $k(\cdot, \cdot)$ (see Eq. (12)) to measure sample similarity. Samples that are closer in feature space generally yield stronger drift, providing richer training signals. This goal is aligned with the motivation of self-supervised learning. A strong feature encoder reduces the occurrence of a nearly "flat" kernel (*i.e.*, $k(\cdot, \cdot)$ vanishes because all samples are far away).

On the other hand, we report that we were unable to make our method work on ImageNet without a feature encoder. In this case, the kernel may fail to effectively describe similarity, even in the presence of a latent VAE. We leave further study of this limitation for future work.

**System-level Comparisons.** In addition to the ablation setting, we train stronger variants and summarize them in Table 4. We compare with previous methods in Table 5.

Our method achieves **1.54** FID with *native* 1-NFE generation. It outperforms all previous 1-NFE methods, which are based on approximating diffusion-/flow-based trajectories. Notably, our Base-size model competes with previous XL-size models. Our best model (FID 1.54) uses a CFG scale of 1.0, which corresponds to "no CFG" in diffusion-based methods. Our CFG formulation exhibits a tradeoff between

*Table 6.* **System-level comparison: ImageNet 256×256 genera-tion in pixel space.** FID is on 50K images, all reported with CFG if applicable. The parameter numbers are of the generator. All generators are trained from scratch (*i.e.*, not distilled).

| method | space | params | NFE | FID↓ | IS↑ |
|---|---|---|---|---|---|
| *Multi-step Diffusion/Flows* | | | | | |
| ADM-G (Dhariwal & Nichol, 2021) | pix | 554M | 250×2 | 4.59 | 186.7 |
| SiD, UViT/2 (Hoogeboom et al., 2023) | pix | 2.5B | 1000×2 | 2.44 | 256.3 |
| VDM++, UViT/2 (Kingma & Gao, 2023) | pix | 2.5B | 256×2 | 2.12 | 267.7 |
| SiD2, UViT/2 (Hoogeboom et al., 2024) | pix | – | 512×2 | 1.73 | – |
| SiD2, UViT/1 (Hoogeboom et al., 2024) | pix | – | 512×2 | **1.38** | – |
| JiT-G/16 (Li & He, 2025) | pix | 2B | 100×2 | 1.82 | 292.6 |
| PixelDiT/16 (Yu et al., 2025) | pix | 797M | 200×2 | 1.61 | 292.7 |
| *Single-step Diffusion/Flows* | | | | | |
| EPG-L/16 (Lei et al., 2025) | pix | 540M | 1 | 8.82 | – |
| *GANs* | | | | | |
| BigGAN (Brock et al., 2018) | pix | 112M | 1 | 6.95 | 152.8 |
| GigaGAN (Kang et al., 2023) | pix | 569M | 1 | 3.45 | 225.5 |
| StyleGAN-XL (Sauer et al., 2022) | pix | 166M | 1 | 2.30 | 265.1 |
| *Drifting Models* | | | | | |
| **Drifting Model, B/16** | pix | 134M | 1 | 2.15 | 287.5 |
| **Drifting Model, L/16** | pix | 464M | 1 | **1.73** | 288.5 |

FID and IS (see B.3), similar to standard CFG.

We provide uncurated qualitative results in Appendix B.5, Fig. 7-10, with CFG 1.0. Moreover, Fig. 11-15 show a side-by-side comparison with improved MeanFlow (iMF) (Geng et al., 2025b), a recent state-of-the-art one-step method.

**Pixel-space Generation.** Our method can naturally work *without* the latent VAE, *i.e.*, the generator $f$ directly produces 256×256×3 images. The feature encoder is applied on the generated images for computing drifting loss. We adopt a configuration similar to that of the latent variant; implementation details are in Appendix A.

Table 6 compares different pixel-space generators. Our *one-step*, *pixel-space* method achieves **1.73** FID, which outperforms or competes with previous multi-step methods. Comparing with other one-step, pixel-space methods (GANs), our method achieves 1.73 FID using only 87G FLOPs; by comparison, StyleGAN-XL produces 2.30 FID using 1574G FLOPs. More ablations are in B.1.

### 5.3. Experiments on Robotic Control

Beyond image generation, we further evaluate our method on robotics control. Our experiment designs and protocols follow *Diffusion Policy* (Chi et al., 2023). At the core of Diffusion Policy is a multi-step, diffusion-based generator; we replace it with our one-step Drifting Model. We directly compute drifting loss on the *raw* representations for control, using no feature space. Results are in Table 7. Our 1-NFE model matches or exceeds the state-of-the-art Diffusion Policy that uses 100 NFE. This comparison suggests that Drifting Models can serve as a promising generative model

*Table 7.* **Robotics Control: Comparison with Diffusion Policy.** The evaluation protocol follows Diffusion Policy (Chi et al., 2023). This table involves four single-stage tasks and two multi-stage tasks. "Drifting Policy" (ours) replaces the multi-step Diffusion Policy generator with our one-step generator. Success rates are reported as the average over the last 10 checkpoints.

| Task | Setting | Diffusion Policy NFE: 100 | **Drifting Policy** **NFE: 1** |
|---|---|---|---|
| *Single-Stage Tasks (State & Visual Observation)* | | | |
| Lift | State | 0.98 | **1.00** |
| | Visual | **1.00** | **1.00** |
| Can | State | 0.96 | **0.98** |
| | Visual | 0.97 | **0.99** |
| ToolHang | State | 0.30 | **0.38** |
| | Visual | **0.73** | 0.67 |
| PushT | State | **0.91** | 0.86 |
| | Visual | 0.84 | **0.86** |
| *Multi-Stage Tasks (State Observation)* | | | |
| BlockPush | Phase 1 | 0.36 | **0.56** |
| | Phase 2 | 0.11 | **0.16** |
| Kitchen | Phase 1 | **1.00** | **1.00** |
| | Phase 2 | **1.00** | **1.00** |
| | Phase 3 | **1.00** | 0.99 |
| | Phase 4 | **0.99** | 0.96 |

across different domains.

## 6. Discussion and Conclusion

We present *Drifting Models*, a new paradigm for generative modeling. At the core of our model is the idea of modeling the evolution of pushforward distributions *during training*. This allows us to focus on the update rule, *i.e.*, $\mathbf{x}_{i+1} = \mathbf{x}_i + \Delta\mathbf{x}_i$, during the iterative training process. This is in contrast with diffusion-/flow-based models, which perform the iterative update at *inference* time. Our method naturally performs one-step inference.

Given that our methodology is substantially different, many open questions remain. For example, although we show that $q = p \Rightarrow \mathbf{V} = \mathbf{0}$, the converse implication does not generally hold in theory. While our designed $\mathbf{V}$ performs well empirically, it remains unclear under what conditions $\mathbf{V} \to \mathbf{0}$ leads to $q \to p$.

From a practical standpoint, although our paper presents an effective instantiation of drifting modeling, many of our design decisions may remain sub-optimal. For example, the design of the drifting field and its kernels, the feature encoder, and the generator architecture remain open for future exploration.

From a broader perspective, our work reframes iterative neural network training as a mechanism for distribution evolution, in contrast to the differential equations underlying diffusion-/flow-based models. We hope that this perspective will inspire the exploration of other realizations of this mechanism in future work.

## Impact Statement

This paper presents work whose goal is to advance the field of machine learning. There are many potential societal consequences of our work, none of which we feel must be specifically highlighted here.

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

# A. Additional Implementation Details

Table 8 summarizes the configurations and hyper-parameters for ablation studies and system-level comparisons. We provide detailed experimental configurations for reproducibility. All ablation studies share a common default setup, while system-level comparisons use scaled-up configurations. More implementation details are described as follows.

## A.1. Pseudo-code for Computing Drifting Field V

Alg. 2 provides the pseudo-code for computing $\mathbf{V}$. The computation is based on taking empirical means in Eq. (11) and (12), which are implemented as softmax over $\mathbf{y}$-sample axis. In practice, we further normalize over the $\mathbf{x}$-sample axis, also implemented as softmax on the same logit matrix. We ablate its influence in B.2.

It is worth noting that this implementation preserves the desired property of $\mathbf{V}$. In principle, this implementation can be viewed as a Monte Carlo estimation of a drifting field:

$$\mathbf{V}_{p,q}(\mathbf{x}) = \mathbb{E}_{\mathcal{B},p,q}[\tilde{K}_{\mathcal{B}}(\mathbf{x},\mathbf{y}^+)\tilde{K}_{\mathcal{B}}(\mathbf{x},\mathbf{y}^-)(\mathbf{y}^+ - \mathbf{y}^-)], \tag{17}$$

where $\mathcal{B}$ consists of other samples in the batch and $\tilde{K}_{\mathcal{B}}$ denote normalizing the distance based on statistics within $\mathcal{B}$. This $\mathbf{V}$ also satisfies $\mathbf{V}_{p,p}(\mathbf{x}) = \mathbf{0}$, since when $p = q$, the term $\tilde{K}_{\mathcal{B}}(\mathbf{y}^+, x)\tilde{K}_{\mathcal{B}}(\mathbf{y}^-, x)(\mathbf{y}^+ - \mathbf{y}^-)$ cancels out with the term $\tilde{K}_{\mathcal{B}}(\mathbf{y}^-, x)\tilde{K}_{\mathcal{B}}(\mathbf{y}^+, x)(\mathbf{y}^- - \mathbf{y}^+)$.

## A.2. Generator Architecture

**Input and output.** The input to the generator consists of random noise along with conditioning:

$$f_\theta : (\boldsymbol{\epsilon}, c, \alpha) \mapsto \mathbf{x}$$

where $\boldsymbol{\epsilon}$ denotes random variables, $c$ is a class label, and $\alpha$ is the CFG strength. $\boldsymbol{\epsilon}$ may consist of both continuous random variables (*e.g.*, Gaussian noise) and discrete ones (*e.g.*, uniformly distributed integers; see random style embeddings). For latent-space models, the output $\mathbf{x} \in \mathbb{R}^{32 \times 32 \times 4}$ is in the SD-VAE latent space. For pixel-space models, the output $\mathbf{x} \in \mathbb{R}^{256 \times 256 \times 3}$ is directly an image.

**Transformer.** We adopt a DiT-style Transformer (Peebles & Xie, 2023). Following (Yao et al., 2025), we use SwiGLU (Shazeer, 2020), RoPE (Su et al., 2024), RMSNorm (Zhang & Sennrich, 2019), and QK-Norm (Henry et al., 2020). The input Gaussian noise is patchified into 256=16×16 tokens (patch size 2×2 for latent, 16×16 for pixel). Conditioning $(c, \alpha)$ is processed by adaLN, as well as by in-context conditioning tokens. The output tokens are unpatchified back to the target shape.

**In-context tokens.** Following (Li & He, 2025), we prepend 16 learnable tokens to the sequence for in-context conditioning (Peebles & Xie, 2023). These tokens are formed by

---

**Algorithm 2** Computing the drifting field $\mathbf{V}$.

```python
def compute_V(x, y_pos, y_neg, T):
  # x: [N, D]
  # y_pos: [N_pos, D]
  # y_neg: [N_neg, D]
  # T: temperature

  # compute pairwise distance
  dist_pos = cdist(x, y_pos) # [N, N_pos]
  dist_neg = cdist(x, y_neg) # [N, N_neg]

  # ignore self (if y_neg is x)
  dist_neg += eye(N) * 1e6

  # compute logits
  logit_pos = -dist_pos / T
  logit_neg = -dist_neg / T

  # concat for normalization
  logit = cat([logit_pos, logit_neg], dim=1)

  # normalize along both dimensions
  A_row = logit.softmax(dim=-1)
  A_col = logit.softmax(dim=-2)
  A = sqrt(A_row * A_col)

  # back to [N, N_pos] and [N, N_neg]
  A_pos, A_neg = split(A, [N_pos,], dim=1)

  # compute the weights
  W_pos = A_pos # [N, N_pos]
  W_neg = A_neg # [N, N_neg]
  W_pos *= A_neg.sum(dim=1,keepdim=True)
  W_neg *= A_pos.sum(dim=1,keepdim=True)

  drift_pos = W_pos @ y_pos # [N_x, D]
  drift_neg = W_neg @ y_neg # [N_x, D]

  V = drift_pos - drift_neg
  return V
```

---

summing the projected conditioning vector with positional embeddings.

**Random style embeddings.** Our framework allows arbitrary noise distributions beyond Gaussians. Inspired by StyleGAN (Sauer et al., 2022), we introduce an additional 32 "style tokens": each of which is a random index into a codebook of 64 learnable embeddings. These are summed and added to the conditioning vector. This does not change the sequence length and introduces negligible overhead in terms of parameters and FLOPs. This table reports the effect of style embeddings on our ablation default:

|  | w/o style | w/ style |
|---|---|---|
| FID | 8.86 | **8.46** |

In contrast to diffusion-/flow-based methods, our method can naturally handle different types of noise or random

*Table 8.* **Configurations for ImageNet 256×256.**

| | ablation default | B/2, latent (Table 5) | L/2, latent (Table 5) | B/16, pixel (Table 6) | L/16, pixel (Table 6) |
|---|---|---|---|---|---|
| ***Generator Architecture*** | | | | | |
| arch | DiT-B/2 | DiT-B/2 | DiT-L/2 | DiT-B/16 | DiT-L/16 |
| input size | 32×32×4 | 32×32×4 | 32×32×4 | 32×32×4 | 32×32×4 |
| patch size | 2×2 | 2×2 | 2×2 | 16×16 | 16×16 |
| hidden dim | 768 | 768 | 1024 | 768 | 1024 |
| depth | 12 | 12 | 24 | 12 | 24 |
| register tokens | 16 | 16 | 16 | 16 | 16 |
| style embedding tokens | 32 | 32 | 32 | 32 | 32 |
| ***Feature Encoder for Drifting Loss*** | | | | | |
| arch | ResNet | ResNet | ResNet | ResNet + ConvNeXt-V2 | ResNet + ConvNeXt-V2 |
| SSL pre-train method | latent-MAE | latent-MAE | latent-MAE | pixel-MAE | pixel-MAE |
| ResNet: input size | 32×32×4 | 32×32×4 | 32×32×4 | 256×256×3 | 256×256×3 |
| ResNet: $conv_1$ stride | 1 | 1 | 1 | 8 | 8 |
| ResNet: base width | 256 | 640 | 640 | 640 | 640 |
| ResNet: block type | | | bottleneck | | |
| ResNet: blocks / stage | | | [3, 4, 6, 3] | | |
| ResNet: size / stage | | | $[32^2, 16^2, 8^2, 4^2]$ | | |
| MAE: masking ratio | | | 50% | | |
| MAE: pre-train epochs | 192 | 1280 | 1280 | 1280 | 640 |
| classification finetune | No | 3k steps | 3k steps | 3k steps | 3k steps |
| ***Generator Optimizer*** | | | | | |
| optimizer | | | AdamW ($\beta_1 = 0.9, \beta_2 = 0.95$) | | |
| learning rate | 2e-4 | 4e-4 | 4e-4 | 2e-4 | 2e-4 |
| weight decay | 0.01 | 0.0 | 0.01 | 0.01 | 0.01 |
| warmup steps | 5k | 10k | 10k | 5k | 10k |
| gradient clip | 2.0 | 2.0 | 2.0 | 2.0 | 2.0 |
| training steps | 30k | 200k | 200k | 100k | 100k |
| training epochs | 100 | 1280 | 1280 | 320 | 320 |
| EMA decay | 0.999 | | {0.999, 0.9995, 0.9998, 0.9999} | | |
| ***Drifting Loss Computation*** | | | | | |
| class labels $N_c$ | 64 | 128 | 128 | 64 | 64 |
| positive samples $N_{pos}$ | 64 | 128 | 64 | 64 | 64 |
| generated samples $N_{neg}$ | 64 | 64 | 64 | 64 | 64 |
| effective batch $B$ ($N_c \times N_{neg}$) | 4096 | 8192 | 8192 | 4096 | 4096 |
| temperatures $\tau$ | | | {0.02, 0.05, 0.2}: one loss per $\tau$, sum all loss terms | | |
| ***CFG Configuration*** | | | | | |
| train: CFG $\alpha$ range | $[1, 4]$ | $[1, 4]$ | $[1, 4]$ | $[1, 4]$ | $[1, 4]$ |
| train: CFG $\alpha$ sampling | $p(\alpha) \propto \alpha^{-3}$ | $p(\alpha) \propto \alpha^{-5}$ | 50%: $\alpha=1$, 50%: $p(\alpha) \propto \alpha^{-3}$ | $p(\alpha) \propto \alpha^{-5}$ | $p(\alpha) \propto \alpha^{-5}$ |
| train: uncond samples $N_{uncond}$ | 16 | 32 | 32 | 32 | 32 |
| inference: CFG $\alpha$ search | | | [1.0, 3.5] | | |

variables. With random style embeddings, the input random variables consist of two parts: (1) Gaussian noise, and (2) discrete indices for style embeddings. Our model $f$ produces the pushforward distribution of their joint distribution.

### A.3. Implementation of ResNet-style MAE

In addition to standard self-supervised learning models (MoCo (He et al., 2020), SimCLR(Chen et al., 2020a)), we develop a customized ResNet-style MAE model as the feature encoder for drifting loss.

**Overview.** Unlike standard MAE (He et al., 2022), which is based on ViT (Dosovitskiy et al., 2021), our MAE trains a convolutional ResNet that provides multi-scale features. For latent-space models, the input and output have dimension 32×32×4; for pixel-space models, the input and output have dimension 256×256×3.

Our MAE consists of a ResNet-style encoder paired with a deconvolutional decoder in a U-Net-style (Ronneberger et al., 2015) encoder-decoder architecture. We only use the ResNet-style encoder for feature extraction when computing the drifting loss.

**MAE Encoder.** The encoder follows a classical ResNet (He et al., 2016) design. It maps an input to multi-scale feature maps (4 scales in ResNet):

$$\text{Encoder} : \mathbf{x} \mapsto \{\mathbf{f}_1, \mathbf{f}_2, \mathbf{f}_3, \mathbf{f}_4\}$$

Here, a feature map $\mathbf{f}_i$ has dimension $H_i \times W_i \times C_i$, with $H_i \times W_i \in \{32^2, 16^2, 8^2, 4^2\}$ and $C_i \in \{C, 2C, 4C, 8C\}$ for a base width $C$.

The architecture follows standard ResNet (He et al., 2016) design, with GroupNorm (GN) (Wu & He, 2018) used in place of BatchNorm (BN) (Ioffe & Szegedy, 2015). All residual blocks are "basic" blocks (*i.e.*, each consisting of two 3×3 convolutions). Following the standard ResNet-34 (He et al., 2016): the encoder has a 3×3 convolution (without downsampling) and 4 stages with [3, 4, 6, 3] blocks; downsampling (stride 2) happens at the first block of stages 2 to 4.

For latent-space (*i.e.*, latent-MAE), the input of this ResNet is $32{\times}32{\times}4$; for pixel-space, the $256{\times}256{\times}3$ input is first patchified (by a $8{\times}8$ patch) into $32{\times}32{\times}192$. The ResNet operates on the input with $H{\times}W = 32{\times}32$.

**MAE Decoder.** The decoder returns to the input shape via deconvolutions and skip connections:

$$\text{Decoder} : \{\mathbf{f}_4, \mathbf{f}_3, \mathbf{f}_2, \mathbf{f}_1\} \mapsto \hat{\mathbf{x}}.$$

It starts with a $3{\times}3$ convolutional block on $\mathbf{f}_4$, followed by 4 upsampling blocks. Each upsampling block performs: bilinear $2{\times}2$ upsampling $\rightarrow$ concatenating with encoder's skip connection $\rightarrow$ GN $\rightarrow$ two $3{\times}3$ convolutions with GN and ReLU. A final $1{\times}1$ convolution produces the output channels. For the pixel-space, the decoder unpatchifies back to the original resolution after the last layer.

**Masking.** The MAE is trained to reconstruct randomly masked inputs. Unlike the ViT-based MAE (He et al., 2022), which removes the masked tokens from the sequence, we simply zero out masked patches. For the input of a shape $H{\times}W = 32{\times}32$ (in either the latent- or pixel-based case), we mask $2{\times}2$ patches by zeroing. Each patch is independently masked with 50% probability.

**MAE training.** We minimize the $\ell_2$ reconstruction loss on the masked regions. We use AdamW (Loshchilov & Hutter, 2019) with learning rate $4{\times}10^{-3}$ and a batch size of 8192. EMA with decay 0.9995 is used. Following (He et al., 2022), we apply random resized crop augmentation to the input (for the latent setting, images are augmented before being passed through the VAE encoder).

**Classification fine-tuning.** For our best feature encoder (last row of Table 3), we fine-tune the MAE model with a linear classifier head. The loss is $0.1\mathcal{L}_{\text{cls}} + 0.9\mathcal{L}_{\text{recon}}$. We fine-tune all parameters in this MAE for 3k iterations.

## A.4. Other Pretrained Feature Encoders

In addition to our customized MAE, we also evaluate other feature encoders for computing the drifting loss.

**MoCo and SimCLR.** We evaluate publicly available self-supervised encoders trained on ImageNet in pixel space: MoCo (He et al., 2020; Chen et al., 2020b) SimCLR (Chen et al., 2020a). We use the ResNet-50 variant. For latent-space generation, we apply the VAE decoder to map generator outputs from latent space ($32{\times}32{\times}4$) to pixel space ($256{\times}256{\times}3$) before feature extraction. Gradients are back-propagated through both the feature extractor and the VAE decoder.

**MAE with ConvNeXt-V2.** In our pixel-space generator, we also investigate ConvNeXt-V2 (Woo et al., 2023) as the feature encoder. We note that ConvNeXt-V2 is a self-supervised pre-trained model using the MAE objective, followed by classification fine-tuning. Like ResNet, ConvNeXt-V2 is a multi-stage architecture.

## A.5. Multi-scale Features for Drifting Loss

Given an image, the feature encoder produces feature maps at multiple scales, with multiple spatial locations per scale. We compute one drifting loss per feature (*e.g.*, per scale and/or per location). Specifically, we compute the kernel, the drift, and the resulting loss independently for each feature. The resulting losses are summed.

For each stage in a ResNet, we extract features from the output of every 2 residual blocks, together with the final output. This yields a set of feature maps, each of shape $H_i{\times}W_i{\times}C_i$. For each feature map, we produce:

(a) $H_i{\times}W_i$ vectors, one per location (each $C_i$-dim);

(b) 1 global mean and 1 global std (each $C_i$-dim);

(c) $\frac{H_i}{2}{\times}\frac{W_i}{2}$ vectors of means and $\frac{H_i}{2}{\times}\frac{W_i}{2}$ vectors of stds (each $C_i$-dim), computed over $2{\times}2$ patches;

(d) $\frac{H_i}{4}{\times}\frac{W_i}{4}$ vectors of means and $\frac{H_i}{4}{\times}\frac{W_i}{4}$ vectors of stds (each $C_i$-dim), computed over $4{\times}4$ patches.

In addition, for the encoder's input ($H_0{\times}W_0{\times}C_0$), we compute the mean of squared values ($x^2$) per channel and obtain a $C_0$-dim vector.

All resulting vectors here are $C_i$-dim. We compute one drifting loss for each of these $C_i$-dim vectors. All these losses, in addition to the vanilla drifting loss without $\phi$, are summed. This table compares the effect of these designs on our ablation default:

|  | (a,b) | (a-c) | (a-d) |
|---|---|---|---|
| FID | 9.58 | 9.10 | **8.46** |

This shows that our method benefits from richer feature sets. We note that once the feature encoder is run, the computational cost of our drifting loss is negligible: computing multi-scale, multi-location losses incurs little overhead compared to computing a single loss.

## A.6. Feature and Drift Normalization

To balance the multiple loss terms from multiple features, we perform normalization for each feature $\phi_j$, where, $\phi_j$ denotes a feature at a specific spatial location within a given scale (see A.5). Intuitively, we want to perform normalization such that the kernel $k(\cdot, \cdot)$ and the drift $\mathbf{V}$ are insensitive to the absolute magnitude of features. This allows our model to robustly support different feature encoders (see Table 3) as well as a rich set of features from one encoder.

**Feature Normalization.** Consider a feature $\phi_j \in \mathbb{R}^{C_j}$. We

define a normalization scale $S_j \in \mathbb{R}^1$ and the normalized feature is denoted by:

$$\tilde{\phi}_j := \phi_j / S_j. \tag{18}$$

When using $\tilde{\phi}_j$, the $\ell_2$ distance computed in Eq. (12) is:

$$dist_j(\mathbf{x}, \mathbf{y}) = \|\tilde{\phi}_j(\mathbf{x}) - \tilde{\phi}_j(\mathbf{y})\|, \tag{19}$$

where $\mathbf{x}$ denotes a generated sample and $\mathbf{y}$ denotes a positive/negative sample, and $\tilde{\phi}_j(\cdot)$ means extracting their feature at $j$. We want the *average* distance to be $\sqrt{C_j}$:

$$\mathrm{E}_{\mathbf{x}}\mathrm{E}_{\mathbf{y}}[dist_j(\mathbf{x}, \mathbf{y})] \approx \sqrt{C_j}. \tag{20}$$

To achieve this, we set the normalization scale $S_j$ as:

$$S_j = \frac{1}{\sqrt{C_j}} \mathrm{E}_{\mathbf{x}}\mathrm{E}_{\mathbf{y}}[\|\phi_j(\mathbf{x}) - \phi_j(\mathbf{y})\|] \tag{21}$$

In practice, we use all $\mathbf{x}$ and $\mathbf{y}$ samples in a batch to compute the empirical mean in place of the expectation. We reuse the `cdist` computation in Alg. 2 for computing the pairwise distances. We apply stop-gradient to $S_j$, because this scalar is conceptually computed from samples from the previous batch.

With the normalized feature, the kernel in Eq. (12) is set as:

$$k(\mathbf{x}, \mathbf{y}) = \exp\left(-\frac{1}{\tilde{\tau}_j}\|\tilde{\phi}_j(\mathbf{x}) - \tilde{\phi}_j(\mathbf{y})\|\right), \tag{22}$$

where $\tilde{\tau}_j := \tau \cdot \sqrt{C_j}$. By doing so, the value of temperature $\tau$ does not depend on the feature magnitude or feature dimensionality. We set $\tau \in \{0.02, 0.05, 0.2\}$ (discussed next).

**Drift Normalization.** When using the feature $\phi_j$, the resulting drift is in the same feature space as $\phi_j$, denoted as $\mathbf{V}_j$. We perform a drift normalization on $\mathbf{V}_j$, for each feature $\phi_j$. Formally, we define a normalization scale $\lambda_j \in \mathbb{R}^1$ and denote:

$$\tilde{\mathbf{V}}_j := \mathbf{V}_j / \lambda_j. \tag{23}$$

Again, we want the normalized drift to be insensitive to the feature magnitude:

$$\mathbb{E}\left[\frac{1}{C_j}\|\tilde{\mathbf{V}}_j\|^2\right] \approx 1. \tag{24}$$

To achieve this, we set $\lambda_j$ as:

$$\lambda_j = \sqrt{\mathbb{E}\left[\frac{1}{C_j}\|\mathbf{V}_j\|^2\right]}. \tag{25}$$

In practice, the expectation is replaced with the empirical mean computed over the entire batch.

With the normalized feature and normalized drift, the drifting loss of the feature $\phi_j$ is:

$$\mathcal{L}_j = \mathrm{MSE}(\tilde{\phi}_j(\mathbf{x}) - \mathrm{sg}(\tilde{\phi}_j(\mathbf{x}) + \tilde{\mathbf{V}}_j)), \tag{26}$$

where MSE denotes mean squared error. The overall loss is the sum across all features: $\mathcal{L} = \sum_j \mathcal{L}_j$.

**Multiple temperatures.** Using normalized feature distances, the value of temperature $\tau$ determines *what is considered "nearby"*. To improve robustness across different features and across different pretrained models we study, we adopt multiple temperatures.

Formally, for each $\tau$ value, we compute the normalized drift as described above, denoted by $\hat{\mathbf{V}}_{j,\tau}$. Then we compute an aggregated field: $\hat{\mathbf{V}}_j \leftarrow \sum_\tau \tilde{\mathbf{V}}_{j,\tau}$, and renormalize it in the same way as Eq. (25): $\hat{\mathbf{V}}_j \leftarrow \hat{\mathbf{V}}_j / \hat{\lambda}_j$. This $\hat{\mathbf{V}}_j$ is used in place of $\tilde{\mathbf{V}}_j$ in (26).

This table shows the effect of multiple temperatures on our ablation default:

| $\tau$ | 0.02 | 0.05 | 0.2 | {0.02, 0.05, 0.2} |
|---|---|---|---|---|
| FID | 10.62 | **8.67** | 8.96 | **8.46** |

Using multiple temperatures can achieve slightly better results than using a single optimal temperature. We fix $\tau \in \{0.02, 0.05, 0.2\}$ and do not require tuning this hyperparameter across different configurations.

**Normalization across spatial locations.** For a feature map of resolution $H_i \times W_i$, there are $H_i \times W_i$ per-location features. Separately computing the normalization for each location would be slow and unnecessary. We assume that features at different locations within the same feature map share the same normalization scale. Accordingly, we concatenate all $H_i \times W_i$ locations and compute the normalization scale over all of them. The feature normalization and drift normalization are both performed in this way.

### A.7. Classifier-Free Guidance (CFG)

To support CFG, at training time, we include $N_{\mathrm{unc}}$ additional unconditional samples (real images from random classes) as extra negatives. These samples are weighted by a factor $w$ when computing the kernel. For a generated sample $\mathbf{x}$, the effective negative distribution it compares with is:

$$\tilde{q}(\cdot|c) \triangleq \frac{(N_{\mathrm{neg}}-1) \cdot q_\theta(\cdot|c) + N_{\mathrm{unc}}w \cdot p_{\mathrm{data}}(\cdot|\varnothing)}{(N_{\mathrm{neg}}-1) + N_{\mathrm{unc}}w}.$$

Comparing this equation with Eq. (15)(16), we have:

$$\gamma = \frac{N_{\mathrm{unc}}w}{(N_{\mathrm{neg}}-1) + N_{\mathrm{unc}}w}$$

and

$$\alpha = \frac{1}{1-\gamma} = \frac{(N_{\mathrm{neg}}-1) + N_{\mathrm{unc}}w}{N_{\mathrm{neg}}-1}.$$

Given a CFG strength $\alpha$, we compute $w$ accordingly, which is used to weight the kernel. The same weighting $w$ is also applied when computing the global distance normalization.

We train our model with CFG-conditioning (Geng et al., 2025b). At each iteration, we randomly sample $\alpha$ following a pre-defined distribution (see Table 8) and compute the resulting $w$ for weighting the unconditional samples. The value of $\alpha$ is a condition input to the network $f_\theta(\boldsymbol{\epsilon}, c, \alpha)$, alongside the class label $c$.

At inference time, we specify a value of $\alpha$. The inference-time computation remains to be one-step (1-NFE).

### A.8. Sample Queue

Our method requires access to randomly sampled *real* (positive/unconditional) data. This can be implemented using a specialized data loader. Instead, we adopt a *sample queue* of cached data, similar to the queue used in MoCo (He et al., 2020). This implementation samples data in a statistically similar way to a specialized data loader. For completeness, we describe our implementation as follows, while noting that a data loader would be a more principled solution.

For each class label, we keep a queue of size 128; for unconditional samples (used in CFG), we maintain a separate global queue of size 1000. At each training step, we push the latest 64 new real (positive/unconditional) samples, alongside their labels, into the corresponding queues; the earliest ones are dequeued. When sampling, positive samples are drawn from the queue of the corresponding class, and unconditional samples are drawn from the global queue. We sample without replacement.

### A.9. Training Loop

In summary, in the training loop, each step proceeds as:

1. Sample a batch ($N_c$) of class labels.

2. For each label $c$, sample a CFG scale $\alpha$.

3. Sample a batch ($N_{\text{neg}}$) of noise $\boldsymbol{\epsilon}$. Feed ($\boldsymbol{\epsilon}, c, \alpha$) to the generator $f$ to produce generated samples;

4. Sample positive samples (same class, $N_{\text{pos}}$) and unconditional samples (for CFG, $N_{\text{unc}}$);

5. Extract features on all generated, positive, and unconditional samples

6. Compute the drifting loss using the features.

7. Run backpropagation and parameter update.

*Table 9.* **Ablations on pixel-space generation.** We study generation directly in pixel space (without VAE). Applying the same MAE recipe as in latent space yields higher FID, indicating that pixel-space generation is more challenging. Combining MAE with ConvNeXt-V2 helps close this gap. Latent-space results shown for reference. The results below follow the ablation setting (B/16 model for pixel-space, 100 epochs).

| feature encoder $\phi$ | FID (100-epoch) | |
| --- | --- | --- |
| | latent (B/2) | pixel (B/16) |
| MAE (width 256, epoch 192) | 8.46 | 32.11 |
| MAE (width 640, epoch 1280) + cls ft. | 3.36 | 9.35 |
| + MAE w/ ConvNeXt-V2 | – | **3.70** |

*Table 10.* **Pixel-space generation: from ablation to final setting.** Beyond the ablation setting, we compare the settings that lead to the results in Table 6.

| case | arch | ep | FID |
| --- | --- | --- | --- |
| (a) baseline (from Table 9) | B/16 | 100 | 3.70 |
| (b) longer | B/16 | 320 | 2.15 |
| (c) larger model | L/16 | 320 | **1.73** |

## B. Additional Experimental Results

### B.1. Ablations on Pixel-Space Generation

We provide more ablations on pixel-space generation in Table 9 and 10. Table 9 compares the effect of the feature encoder on the pixel-space generator. It shows that the choice of feature encoder plays a more significant role in pixel-space generation quality. A weaker MAE encoder yields an FID of 32.11, whereas a stronger MAE encoder improves performance to an FID of 9.35. We further add another feature encoder, ConvNeXt-V2 (Woo et al., 2023), which is also pre-trained with the MAE objective. This further improves the result to an FID of 3.70.

Table 10 reports the results of training longer and using a larger model. Due to limited time, we train pixel-space models for only 320 epochs; we expect that longer training would yield further improvements. We achieve an FID of 1.73 for pixel-space generation. This is our result in the main paper (Table 6).

### B.2. Ablation on Kernel Normalization

In Eq. (11), our drifting field is weighted by normalized kernels, which can be written as:

$$\mathbf{V}(\mathbf{x}) = \mathbb{E}_{p,q}[\tilde{k}(\mathbf{x}, \mathbf{y}^+)\tilde{k}(\mathbf{x}, \mathbf{y}^-)(\mathbf{y}^+ - \mathbf{y}^-)], \quad (27)$$

where $\tilde{k}(\cdot, \cdot) = \frac{1}{Z}k(\cdot, \cdot)$ denotes the normalized kernel. In principle, this normalization is approximated by a softmax operation over the axis of $\mathbf{y}$ samples. Our implementation (Alg. 2) further applies softmax over the axis of $\mathbf{x}$ samples. We compare these designs, along with another variant without normalization ($Z = 1$).

*Table 11.* **Ablation on kernel normalization.** Softmax normalization over both the **x** and **y** axes performs better. On the other hand, even using no normalization performs decently, showing the robustness of our method. (Setting: B/2 model, 100 epochs)

| kernel normalization | FID |
| --- | --- |
| softmax over **x** and **y** (default) | **8.46** |
| softmax over **y** | 8.92 |
| no normalization | 10.54 |

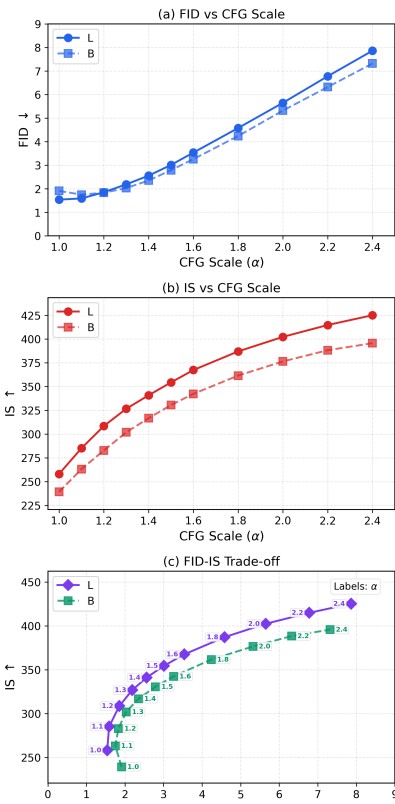

*Figure 5.* **Effect of CFG scale $\alpha$.** **(a)**: FID vs. $\alpha$. **(b)**: IS vs. $\alpha$. **(c)**: IS vs. FID. We show the L/2 (solid) and B/2 (dashed) models. Consistent with common observations in diffusion-/flow-based models, the CFG scale effectively trades off distributional coverage (as reflected by FID) against per-image quality (measured by IS). Notably, with the L/2 model, the optimal FID is achieved at $\alpha$=1.0, which is often regarded as "w/o CFG" in diffusion-/flow-based models. For B/2, the optimal FID is achieved at $\alpha$=1.1.

Table 11 compares the three designs. Using the **y**-only softmax performs well (8.92 FID), whereas using both **x** and **y** softmax improves the result (8.46 FID). On the other hand, even without normalization, performance remains decent, demonstrating the robustness of our method.

We note that all three variants satisfy the equilibrium condition $\mathbf{V}_{p,q}(\mathbf{x}) = \mathbf{0}$ when $p = q$. This explains why all variants perform reasonably well and why even the destructive setting (no normalization) avoids catastrophic failure.

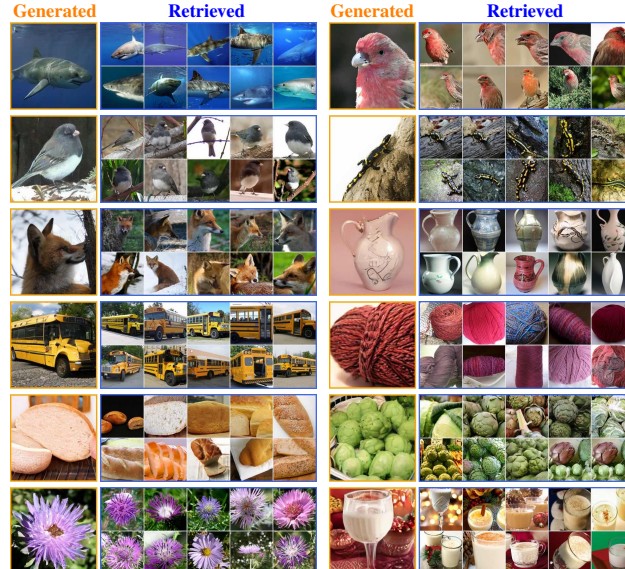

*Figure 6.* **Nearest neighbor analysis.** Each panel shows a generated sample together with its top-10 nearest real images. The nearest neighbors are retrieved from the ImageNet training set based on the cosine similarity using a CLIP encoder (Radford et al., 2021). Our method generates novel images that are visually distinct from their nearest neighbors.

### B.3. Ablation on CFG

In Figure 5, we investigate the CFG scale $\alpha$ used at inference time. It shows that the CFG formulation developed for our models exhibits behavior similar to that observed in diffusion-/flow-based models. Increasing the CFG scale leads to higher IS values, whereas beyond the FID sweet spot, further increases in IS come at the cost of worse FID.

Notably, with our best model (L/2), the optimal FID is achieved at $\alpha$=1.0, which is often regarded as "w/o CFG" in diffusion-/flow-based models (even though their "w/o CFG" setting can reduce NFE by half). While our method need not run an unconditional model at inference time (in contrast to standard CFG), training is influenced by the use of unconditional real samples as negatives.

### B.4. Nearest Neighbor Analysis

In Figure 6, we show generated images together with their nearest real images. The nearest neighbors are retrieved from the ImageNet training set using CLIP features. These visualizations suggest that our method generates novel images that are visually distinct from their nearest neighbors, rather than merely memorizing training samples.

### B.5. Qualitative Results

Fig. 7-10 show uncurated samples from our model. Fig. 11-15 provide side-by-side comparison with improved Mean-Flow (iMF) (Geng et al., 2025b), the current state-of-the-art one-step method.

# C. On Identifiability of the Zero-Drift Equilibrium

**Problem statement.** In Sec. 3, we showed that anti-symmetry implies $p = q \Rightarrow \mathbf{V}(\mathbf{x}) \equiv \mathbf{0}$. Here we investigate the converse: under what conditions does $\mathbf{V}(\mathbf{x}) \approx \mathbf{0}$ imply $p \approx q$? Generally, this is not guaranteed for arbitrary vector fields. However, we argue that for our specific construction, the zero-drift condition imposes strong constraints on the distributions.

**Full-support assumption.** To avoid boundary issues, we assume that $p$ and $q$ have full support on $\mathbb{R}^d$ (e.g., via infinitesimal Gaussian smoothing). Consequently, ensuring the equilibrium condition $\mathbf{V}(\mathbf{x}) \approx \mathbf{0}$ for generated samples $\mathbf{x} \sim q$ effectively enforces $\mathbf{V}(\mathbf{x}) \approx \mathbf{0}$ for all $\mathbf{x} \in \mathbb{R}^d$.

## C.1. A Generic Identifiability Heuristic for General $K$

**Setup.** Consider a general interaction kernel $K(\mathbf{x}, \mathbf{y}^+, \mathbf{y}^-) \in \mathbb{R}^d$ and the drifting field

$$\mathbf{V}_{p,q}(\mathbf{x}) := \mathbb{E}_{\mathbf{y}^+ \sim p, \, \mathbf{y}^- \sim q}\left[K(\mathbf{x}, \mathbf{y}^+, \mathbf{y}^-)\right]. \qquad (28)$$

We assume that $p$ and $q$ belong to a finite-dimensional model class spanned by a linearly independent basis $\{\varphi_i\}_{i=1}^m$:

$$p(\mathbf{y}) = \sum_{i=1}^m a_i \, \varphi_i(\mathbf{y}), \qquad q(\mathbf{y}) = \sum_{i=1}^m b_i \, \varphi_i(\mathbf{y}), \qquad (29)$$

where $\mathbf{a}, \mathbf{b} \in \mathbb{R}^m$ are coefficient vectors.

**Bilinear expansion over test locations.** Consider a set of test locations (probes) $\mathcal{X} = \{\mathbf{x}_k\}_{k=1}^N$ with sufficiently large $N$ (e.g., $N \gg m^2$). For each pair of basis indices $(i, j)$, we define the *induced interaction vector* $\mathbf{U}_{ij} \in \mathbb{R}^{d \times N}$ by computing its column:

$$\mathbf{U}_{ij}[:, \mathbf{x}] \triangleq \iint K(\mathbf{x}, \mathbf{y}^+, \mathbf{y}^-) \, \varphi_i(\mathbf{y}^+) \, \varphi_j(\mathbf{y}^-) \, d\mathbf{y}^+ d\mathbf{y}^- \tag{30}$$

evaluated at all $\mathbf{x} \in \mathcal{X}$. Substituting the basis expansion into Eq. (28), the drifting field evaluated on $\mathcal{X}$ (stored as a matrix $\mathbf{V}_{\mathcal{X}}$) is a bilinear combination:

$$\mathbf{V}_{\mathcal{X}} \triangleq \sum_{i=1}^m \sum_{j=1}^m a_i b_j \mathbf{U}_{ij}. \qquad (31)$$

Here, $\mathbf{V}_{\mathcal{X}} \in \mathbb{R}^{d \times N}$. At the equilibrium, we have $\mathbf{V}_{\mathcal{X}} = \mathbf{0}$, which yields $dN$ linear equations.

**Linear independence assumption.** Our anti-symmetry condition implies that switching $p$ and $q$ negates the field. In terms of basis interactions, this means $\mathbf{U}_{ij} = -\mathbf{U}_{ji}$ (and consequently $\mathbf{U}_{ii} = \mathbf{0}$). We make the *generic non-degeneracy assumption*:

*The set of vectors $\{\mathbf{U}_{ij}\}_{1 \leq i < j \leq m}$ is linearly independent in $\mathbb{R}^{dN}$.*

This assumption requires the probes $\mathcal{X}$ and kernel $K$ to be non-degenerate; if all $\mathbf{x}$ yield identical constraints, independence would fail. For generic choices of $K$ and sufficiently diverse probes $\mathcal{X}$ with $dN \gg m^2$, such linear independence is a natural non-degeneracy condition.

**Uniqueness of the equilibrium.** The zero-drift condition $\mathbf{V}(\mathbf{x}) \equiv \mathbf{0}$ implies $\mathbf{V}_{\mathcal{X}} = \mathbf{0}$. Grouping terms by the independent basis vectors $\{\mathbf{U}_{ij}\}_{i<j}$, we have:

$$\sum_{1 \leq i < j \leq m} (a_i b_j - a_j b_i)\mathbf{U}_{ij} = \mathbf{0}. \qquad (32)$$

By the linear independence assumption, the coefficients must vanish: $a_i b_j - a_j b_i = 0$ for all $i, j$. This implies that the vector $\mathbf{a}$ is parallel to $\mathbf{b}$ (i.e., $\mathbf{a} \propto \mathbf{b}$). Since $p$ and $q$ are probability densities (implying $\int p = \int q = 1$), we must have $\mathbf{a} = \mathbf{b}$, and thus $p = q$.

**Connection to the mean shift field.** The mean-shift field fits this framework. The update vector (before normalization) is $\mathbb{E}_{p,q}[k(\mathbf{x}, \mathbf{y}^+)k(\mathbf{x}, \mathbf{y}^-)(\mathbf{y}^+ - \mathbf{y}^-)]$. Assuming the normalization factors $Z_p$ and $Z_q$ are finite, the condition $\mathbf{V}(\mathbf{x}) = \mathbf{0}$ implies the numerator integral vanishes, which corresponds to an interaction kernel of the form:

$$K(\mathbf{x}, \mathbf{y}^+, \mathbf{y}^-) = k(\mathbf{x}, \mathbf{y}^+) \, k(\mathbf{x}, \mathbf{y}^-) \, (\mathbf{y}^+ - \mathbf{y}^-). \qquad (33)$$

This kernel generates the bilinear structure analyzed above. Since we can choose $N$ such that $dN \gg m^2$, the dimension of the test space is much larger than the number of basis pairs. Thus, the linear independence of $\{\mathbf{U}_{ij}\}$ is expected to hold for generic configurations. Finally, for general distributions $p$ and $q$, we can approximate them using a sufficiently large basis expansion, turning into $\tilde{p}$ and $\tilde{q}$. When the basis approximation is sufficiently accurate, $\tilde{p} \approx p$ and $\tilde{q} \approx q$, and the drift field $\mathbf{V}_{\tilde{p},\tilde{q}} \approx \mathbf{V}_{p,q} \approx 0$. By the argument above, $\tilde{p} \approx \tilde{q}$, and thus $p \approx q$.

The argument above works for general form of drifting field, under mild anti-degeneracy assumptions.

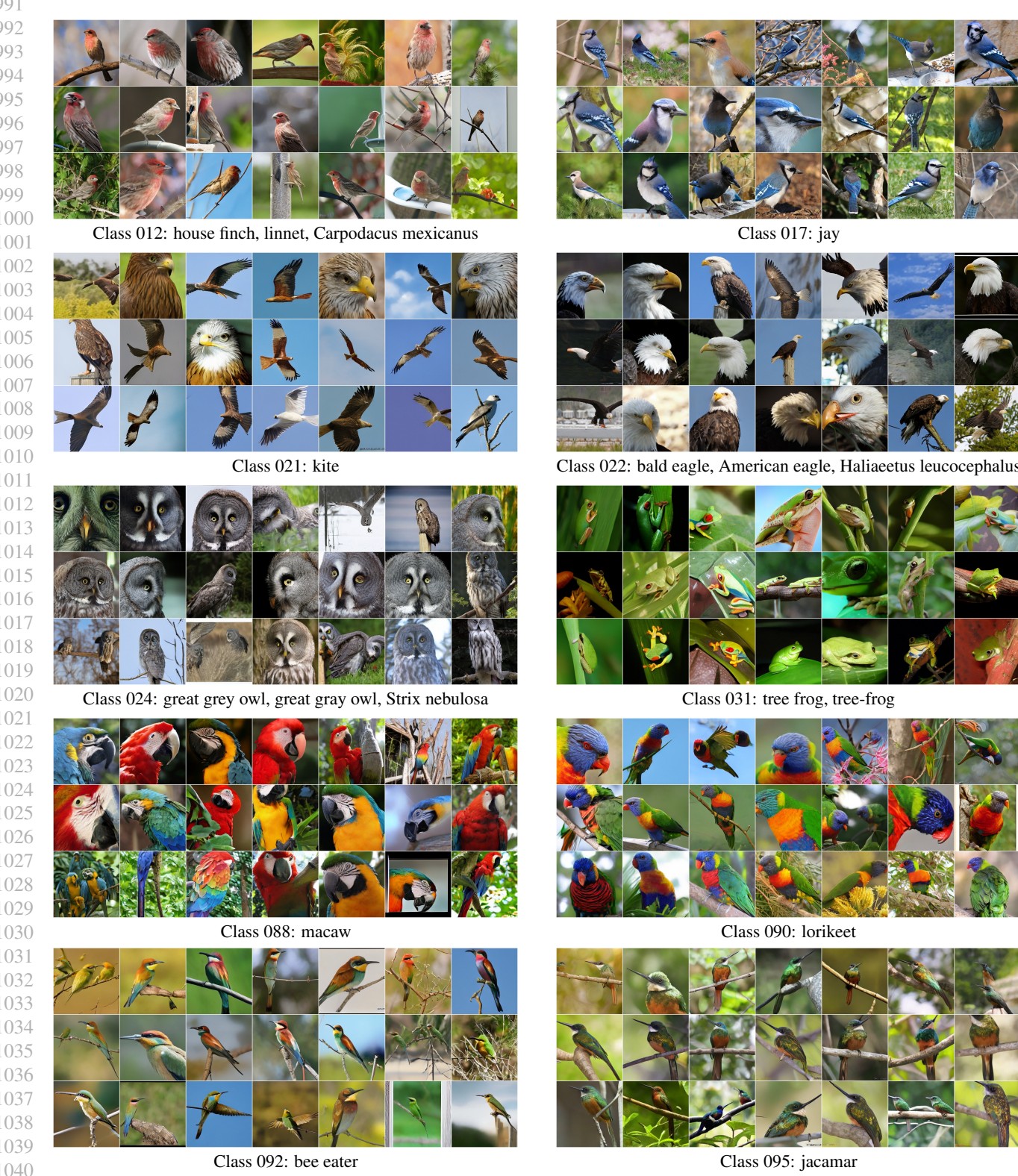

*Figure 7.* **Uncurated samples from our latent-L/2 model with CFG = 1.0 (page 1/4).** FID = 1.54, IS = 258.9.

Class 108: sea anemone, anemone

Class 145: king penguin, Aptenodytes patagonica

Class 270: white wolf, Arctic wolf, Canis lupus tundrarum

Class 279: Arctic fox, white fox, Alopex lagopus

Class 288: leopard, Panthera pardus

Class 291: lion, king of beasts, Panthera leo

Class 296: ice bear

Class 323: monarch, Danaus plexippus

Class 349: bighorn, bighorn sheep, Ovis canadensis

Class 386: African elephant, Loxodonta africana

*Figure 8.* **Uncurated samples from our latent-L/2 model with CFG = 1.0 (page 2/4).** FID = 1.54, IS = 258.9.

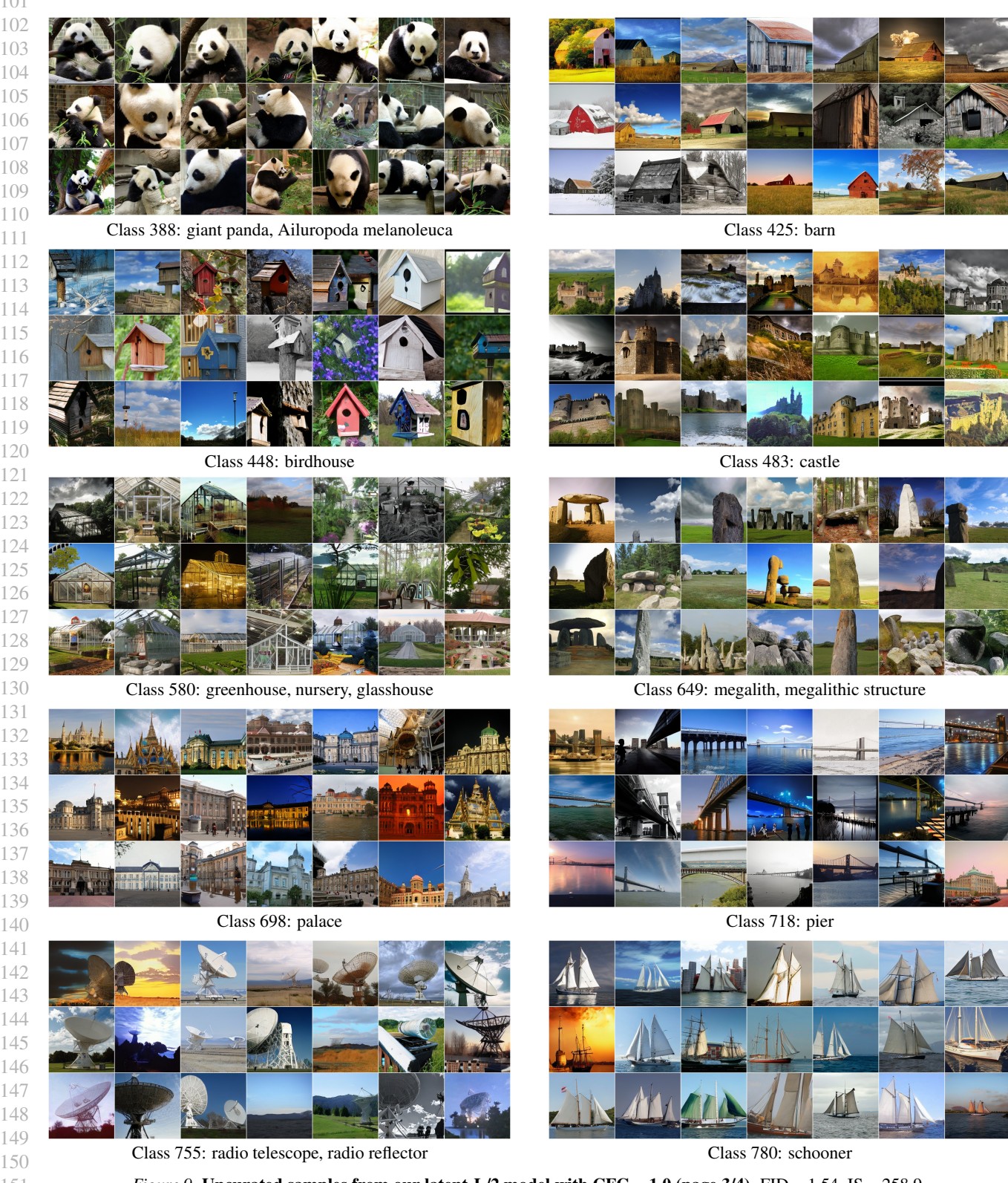

Class 388: giant panda, Ailuropoda melanoleuca

Class 425: barn

Class 448: birdhouse

Class 483: castle

Class 580: greenhouse, nursery, glasshouse

Class 649: megalith, megalithic structure

Class 698: palace

Class 718: pier

Class 755: radio telescope, radio reflector

Class 780: schooner

*Figure 9.* **Uncurated samples from our latent-L/2 model with CFG = 1.0 (page 3/4).** FID = 1.54, IS = 258.9.

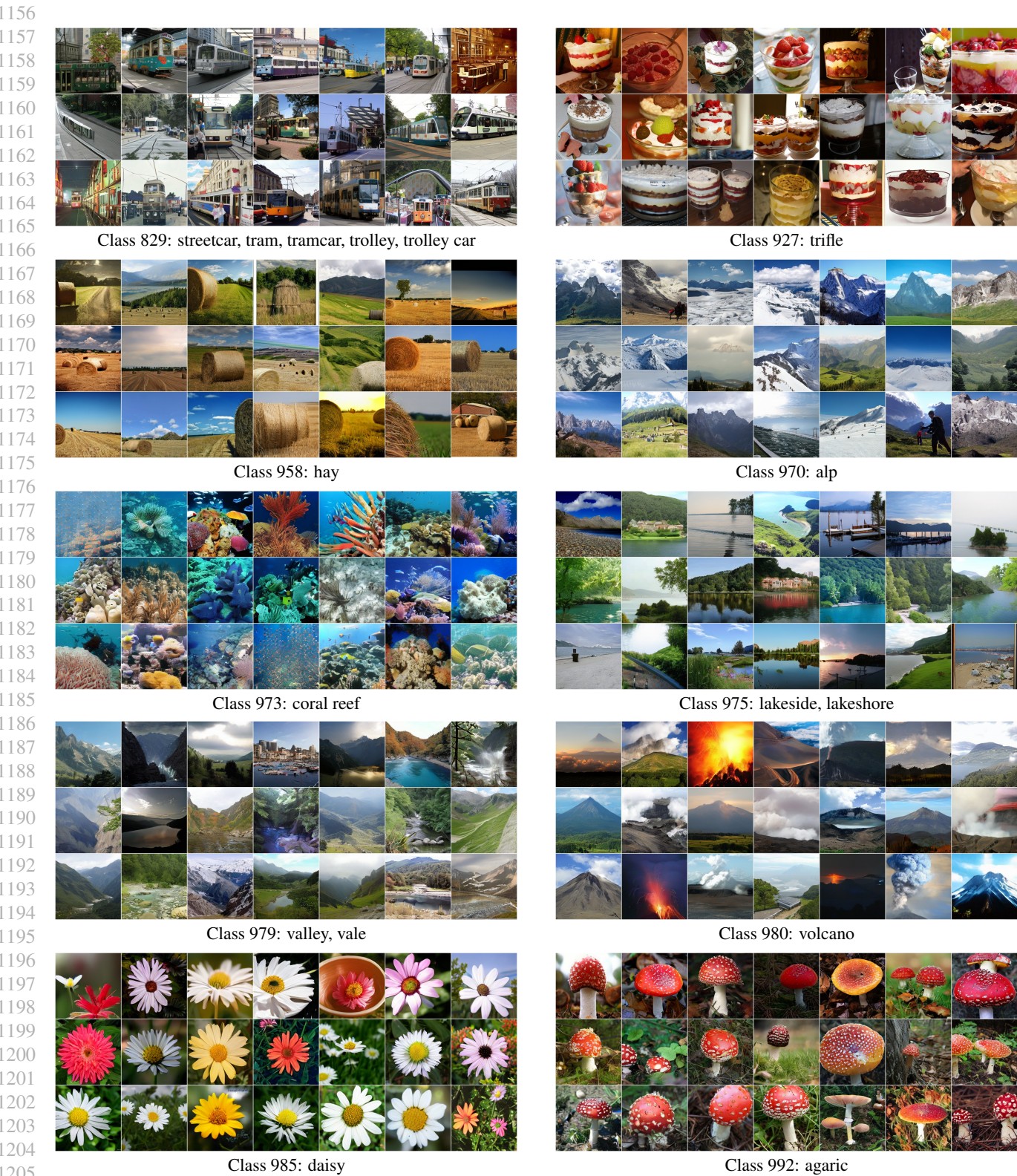

Class 829: streetcar, tram, tramcar, trolley, trolley car

Class 927: trifle

Class 958: hay

Class 970: alp

Class 973: coral reef

Class 975: lakeside, lakeshore

Class 979: valley, vale

Class 980: volcano

Class 985: daisy

Class 992: agaric

*Figure 10.* **Uncurated samples from our latent-L/2 model with CFG = 1.0 (page 4/4).** FID = 1.54, IS = 258.9.

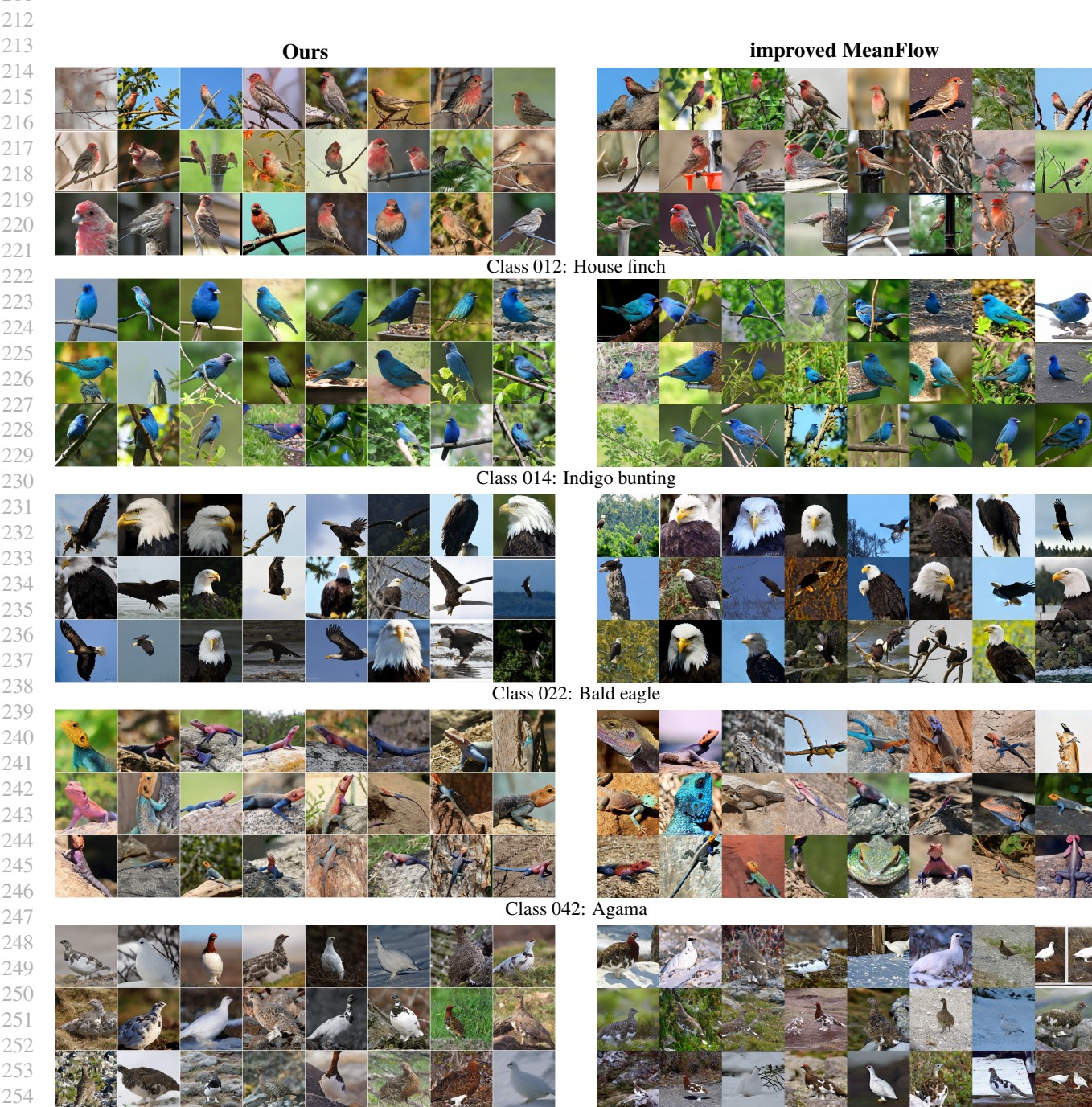

Class 012: House finch

Class 014: Indigo bunting

Class 022: Bald eagle

Class 042: Agama

Class 081: Ptarmigan

*Figure 11.* **Side-by-side comparison with improved MeanFlow (iMF) (Geng et al., 2025b) (page 1/5).** Uncurated samples from our method (left) and iMF (right) on **all** ImageNet classes visualized in the iMF paper. Both methods generate images with a single neural function evaluation (1-NFE). The iMF visualizations use CFG $\omega$=6.0 and interval $[t_{\min}, t_{\max}]$=$[0.2, 0.8]$, achieving FID 3.92 and IS 348.2 (DiT-XL/2). For fair comparison, we set the CFG scale to match the IS of iMF visualizations, which leads to FID 3.01 and IS 354.4 (at CFG=1.5) for our method (DiT-L/2).

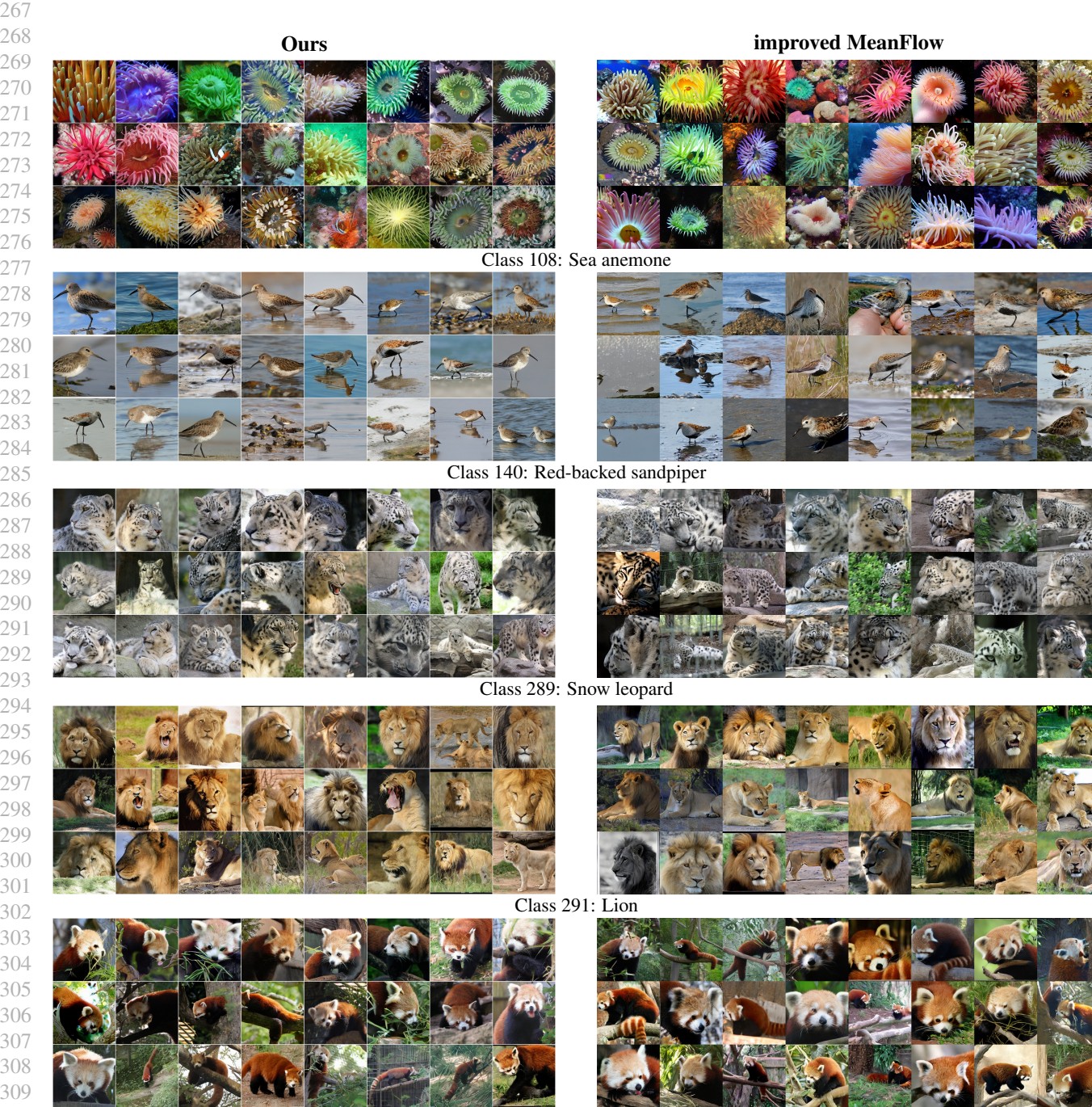

Class 108: Sea anemone

Class 140: Red-backed sandpiper

Class 289: Snow leopard

Class 291: Lion

Class 387: Lesser panda

*Figure 12.* **Side-by-side comparison with improved MeanFlow (iMF) (Geng et al., 2025b) (page 2/5).** Uncurated samples from our method (left) and iMF (right) on **all** ImageNet classes visualized in the iMF paper. Both methods generate images with a single neural function evaluation (1-NFE). The iMF visualizations use CFG $\omega$=6.0 and interval $[t_{min}, t_{max}]$=$[0.2, 0.8]$, achieving FID 3.92 and IS 348.2 (DiT-XL/2). For fair comparison, we set the CFG scale to match the IS of iMF visualizations, which leads to FID 3.01 and IS 354.4 (at CFG=1.5) for our method (DiT-L/2).

**Ours**          **improved MeanFlow**

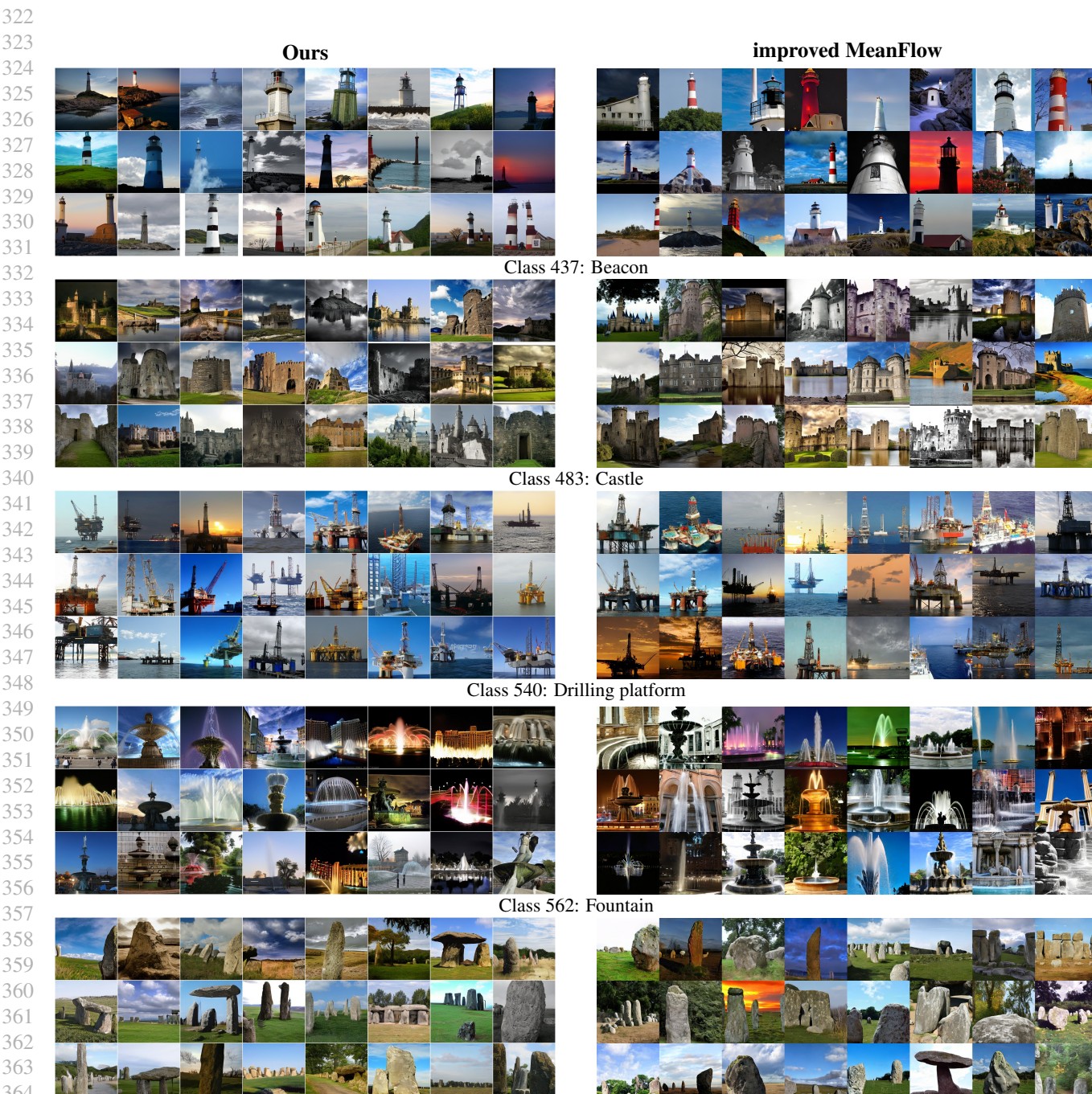

Class 437: Beacon

Class 483: Castle

Class 540: Drilling platform

Class 562: Fountain

Class 649: Megalith

*Figure 13.* **Side-by-side comparison with improved MeanFlow (iMF) (Geng et al., 2025b) (page 3/5).** Uncurated samples from our method (left) and iMF (right) on **all** ImageNet classes visualized in the iMF paper. Both methods generate images with a single neural function evaluation (1-NFE). The iMF visualizations use CFG $\omega$=6.0 and interval $[t_{\min}, t_{\max}]$=[0.2, 0.8], achieving FID 3.92 and IS 348.2 (DiT-XL/2). For fair comparison, we set the CFG scale to match the IS of iMF visualizations, which leads to FID 3.01 and IS 354.4 (at CFG=1.5) for our method (DiT-L/2).

**Ours**                **improved MeanFlow**

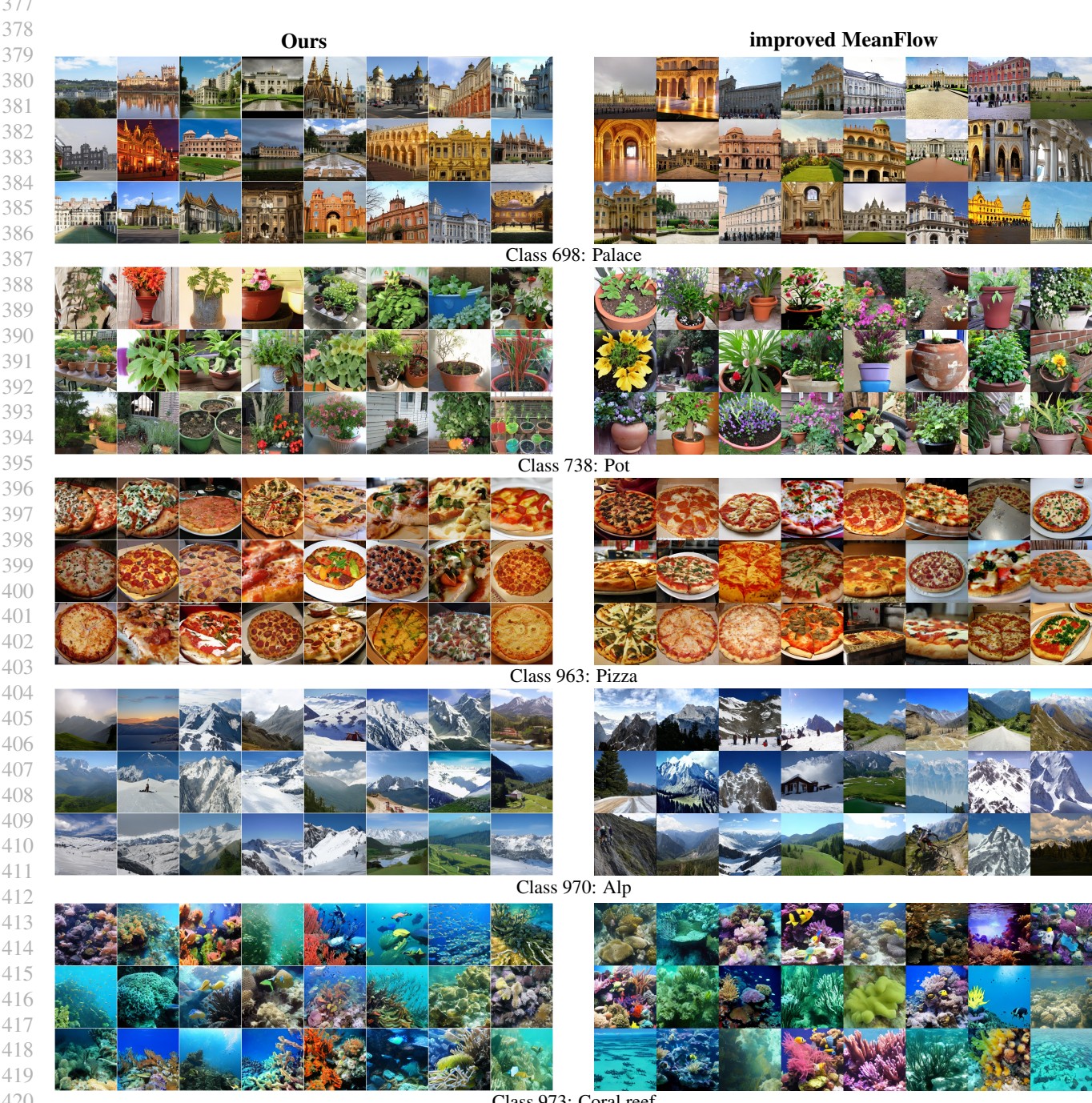

Class 698: Palace

Class 738: Pot

Class 963: Pizza

Class 970: Alp

Class 973: Coral reef

*Figure 14.* **Side-by-side comparison with improved MeanFlow (iMF) (Geng et al., 2025b) (page 4/5).** Uncurated samples from our method (left) and iMF (right) on **all** ImageNet classes visualized in the iMF paper. Both methods generate images with a single neural function evaluation (1-NFE). The iMF visualizations use CFG $\omega$=6.0 and interval $[t_{\min}, t_{\max}]$=$[0.2, 0.8]$, achieving FID 3.92 and IS 348.2 (DiT-XL/2). For fair comparison, we set the CFG scale to match the IS of iMF visualizations, which leads to FID 3.01 and IS 354.4 (at CFG=1.5) for our method (DiT-L/2).

**Ours**                                                  **improved MeanFlow**

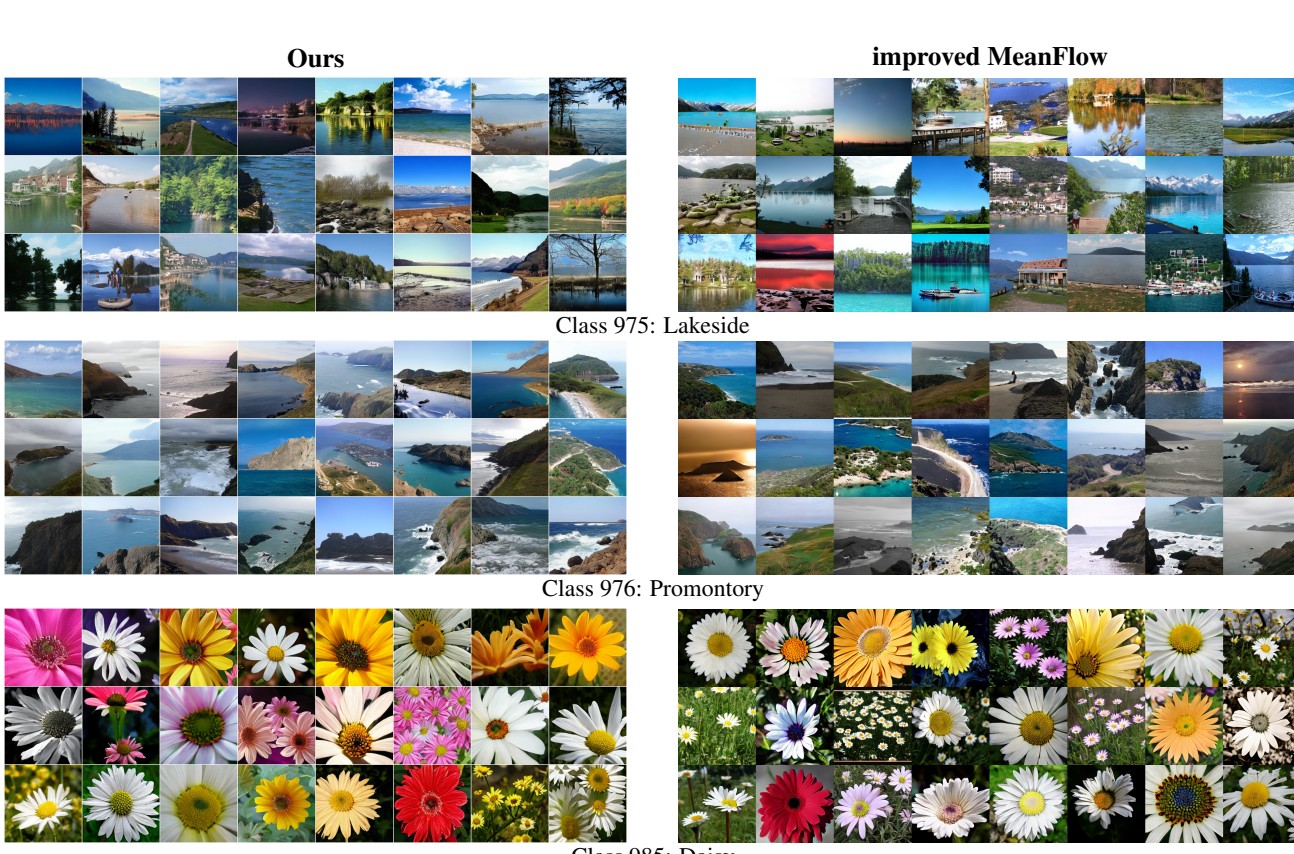

Class 975: Lakeside

Class 976: Promontory

Class 985: Daisy

*Figure 15.* **Side-by-side comparison with improved MeanFlow (iMF) (Geng et al., 2025b) (page 5/5).** Uncurated samples from our method (left) and iMF (right) on **all** ImageNet classes visualized in the iMF paper. Both methods generate images with a single neural function evaluation (1-NFE). The iMF visualizations use CFG $\omega=6.0$ and interval $[t_{\min}, t_{\max}]=[0.2, 0.8]$, achieving FID 3.92 and IS 348.2 (DiT-XL/2). For fair comparison, we set the CFG scale to match the IS of iMF visualizations, which leads to FID 3.01 and IS 354.4 (at CFG=1.5) for our method (DiT-L/2).

