# OpenReview forum: "Generative Modeling via Drifting"
_ICML.cc/2026/Conference — Submitted to ICML 2026_

### Official Review · Reviewer_dwNu · 2026-02-25

**Soundness:** 2
**Presentation:** 3
**Significance:** 3
**Originality:** 3
**Overall Recommendation:** 3
**Confidence:** 4

**Summary:**

The authors propose a new paradigm for one-step generative modeling, dubbed Drifting Models. The idea draws inspiration from contrastive learning literature: it centers around a drifting field that evolves during training as a function of the ground truth data (positives) and learned distribution, $q$, (negatives) such that at optimality the drift vanishes. The proposed method is validated on ImageNet and robotics control (as substitute for a Diffusion Policy). For the former, the authors claim state-of-the-art results in terms of FID.

**Compliance With Llm Reviewing Policy:**

Affirmed.

**Final Justification:**

The authors present and thoroughly benchmark an interesting idea. The paper is well-written and the authors are reasonably honest about the limitations of their work. My remaining concerns are on theoretical understanding of the method / its shortcomings and on implementation / method tuning. These still seem like highly non-trivial issues to me and were also pointed out by other reviewers.

Overall, this is a strong submission with clear merits. However, given the limitations, I find the central claim of a "new generative modeling paradigm" too strong.

**Key Questions For Authors:**

1. One fundamental question is regarding the validity of the proposed method as a generative modeling paradigm. Perhaps a good way to address W1 would be to demonstrate the method in tractable scenarios, amenable to theoretical analysis. For example, can the proposed Drifting Models learn a multivariate Gaussian distribution? Some more insight into identifiability, as in Appendix C, might explain the authors' failed experiments on ImageNet. It would also be useful if the authors share their findings regarding these encoder-free ImageNet Drifting Models.

2. Regarding W2: the hyperparameters in the paper were calibrated for ImageNet. How sensitive is your method to these when applied on other datasets, e.g., faces? Given a fixed compute budget, do Drifting Models outperfom the competition with minimal tuning? Do the authors have general recommendations for stable training of Drifting Models?

3. State-of-the-art FID alone does not imply good generative modeling. A quantitative analysis of similarity to the training data would be helpful (Figure 6 in the Appendix is qualitative). For example, are the generated images significantly more similar to the training set vs some unseen validation set drawn from the same distribution?

Overall, I think this paper demonstrates strong engineering but the scientific contribution should be improved to support the claim of a new generative modeling paradigm.

**Limitations:**

Some limitations are discussed in the paper (e.g., see my W1) and the clarity is appreciated. I encourage the authors to explicitly include a Limitations section in the final version of the paper, some potential examples are listed above in my review.

**Strengths And Weaknesses:**

Strengths
-

1. The paper is well-written and easy to read.
2. The method is shown to be competitive in fairly large scale settings, demonstrating relevance and applicability to modern generative modeling.

Weaknesses
-

1. I am not convinced regarding the theoretical framework. The authors acknowledge in line 369, column 2 that their method fails without using a pretrained feature encoder. Given this, the claimed state-of-art performance is questionable: is it fair to compare this method with ones that do not rely on pretraining? How applicable might Drifting Models be when such specialized pretraining is unavailable?

2. The complexity of this method at training time appears to be significant. For example, as with existing contrastive methods, Drifting Models require a queue and to carefully calibrate the hyperparameters associated with it.

---

> ### Author Rebuttal · Authors · 2026-03-30
>
> ### Response to Reviewer dwNu
>
> We thank Reviewer dwNu for the detailed and thought-provoking feedback. We are glad the reviewer finds the paper well-written and the method competitive at fairly large scales. The questions are especially helpful for improving the scientific framing, particularly on (1) pretrained features, (2) identifiability, (3) training complexity, and (4) memorization. We address each point below.
>
> ---
>
> ### W1 / Q1. Dependence on pretrained feature encoder, identifiability, and tractable settings
>
> We agree that the dependence on pretrained features is a limitation of our current ImageNet instantiation. We also agree that tractable settings and identifiability analysis are important for clarifying the scientific contribution.
>
> **Encoder dependence.** The drifting formulation itself does not require a feature encoder: our toy experiments (checkerboard, swiss roll) and robotics experiments work in raw space without pretrained features. However, for ImageNet experiments, we found FID decreases only very slowly in the encoder-free setting. Our interpretation is that in this regime, high-dimensional distances appear to provide too weak a geometric signal for the kernel, whereas pretrained features provide a more informative space for drift estimation. We plan to make this limitation and its practical role clearer in the revision.
>
> **Feature-space drifting.** Defining drifting fields in feature spaces preserves the same zero-drift logic and supports principled aggregation across multiple feature spaces. We view this as a useful property of the formulation, and Table 3 suggests that performance improves with stronger feature spaces, suggesting an interesting scaling-like trend. At the same time, we agree it is also a practical limitation of the current ImageNet instantiation, since the method remains challenging to train directly in raw space.
>
> **Identifiability theory.** Inspired by the reviewer's suggestion, we investigated tractable scenarios during the rebuttal period and obtained two theoretical results that we plan to include.
>
> 1. **Gaussian kernel (general distributions).** For Gaussian kernel $k(x,y)=e^{-|x-y|^2/2\sigma^2}$, the mean-shift drifting field satisfies
>    $$
>    V_{p,q}(x)=\sigma^2\left[\nabla_x \log \hat p_\sigma(x)-\nabla_x \log \hat q_\sigma(x)\right],
>    $$
>    where $\hat p_\sigma,\hat q_\sigma$ are Gaussian-smoothed densities. If $V_{p,q}(x)=0$ for all $x$, then $\hat p_\sigma=\hat q_\sigma$; as Gaussian convolution is injective, this implies $p=q$. Thus identifiability holds in this setting.
>
> 2. **Laplacian kernel, mean-shift, with Gaussian $P,Q$.** For the practical kernel $k(x,y)=\exp(-\lVert x-y\rVert/\tau)$ with $P=\mathcal N(\mu_p,\Sigma_p)$ and $Q=\mathcal N(\mu_q,\Sigma_q)$, if $V_{p,q}(x)=0$ for all $x$, consider a unit vector $u$ and a scalar $r$. We have:
>    $$
>    V_{p,q}(ru)\to (\mu_p-\mu_q)+(\Sigma_p-\Sigma_q)u/\tau
>    $$
>    as $r\to\infty$. Since $u$ can be any unit vector, this implies identifiability in the multivariate Gaussian case.
>
> The general case remains open.
>
> ---
>
> ### W2 / Q2. Training complexity and hyperparameter sensitivity
>
> We appreciate this concern. We agree that the system is nontrivial. In practice the recipe is simpler than it may appear:
>
> 1. For ImageNet, the queue stores real data samples rather than evolving features, making it much less sensitive than contrastive queues; empirically, a range of reasonable update frequencies worked similarly well.
>
> 2. Across toys, robotics, and ImageNet, the same temperature mixture [0.02, 0.05, 0.2] and standard LR warmup were sufficient.
>
> 3. We also tested the same recipe on unconditional CelebA-64, using the same ImageNet-trained features, and reached 10k-FID 7.6 without task-specific tuning. While not a controlled comparison to prior one-step models, we view this as a transfer evidence beyond ImageNet.
>
> 4. As Table 2 suggests, larger positive/negative sample counts were consistently better, so we use the largest that fit in memory, with temperatures [0.02, 0.05, 0.2] and standard LR tuning.
>
>
> ---
>
> ### W3 / Q3. Memorization and similarity to training data
>
> This is an important question. In addition to the qualitative nearest-neighbor analysis, we performed a quantitative memorization analysis using nearest-neighbor CLIP cosine similarity against the ImageNet training set (1.28M):
>
> | Query set | Reference set | Avg. nearest CLIP sim. |
> |-----------|---------------|------------------------|
> | Generated samples | Training set | 0.850 |
> | SiT XL/2 samples (CFG 1.0) | Training Set | 0.845 |
> | SiT XL/2 samples (CFG 1.5) | Training Set | 0.870 |
> | Validation set | Training set | 0.890 |
>
> The generator's nearest-neighbor similarity is *lower* than that of the held-out validation set, and comparable to diffusion models. This suggests the model is not simply memorizing training images, since generated samples are on average more distant from the training set than real held-out samples.

---

> > ### Author Rebuttal · Reviewer_dwNu · 2026-04-03
> >
> > Thank you for your answers. I also read the other reviews and I am curious about the point Reviewer `XgXL` made regarding high-dimensional settings.
> >
> > > In raw pixel/latent space, high-dimensional distances concentrate and the kernel becomes almost uniform. So the feature encoder isn't a nice-to-have; it's what makes the whole approach viable.
> >
> > It would be helpful if the authors could comment if this is what they observe in practice for their failed experiments. If yes, perhaps heavy tailed kernels might overcome the issue?

---

> > > ### Author Response · Authors · 2026-04-05
> > >
> > > We thank the reviewer for this insightful follow-up. It helped us sharpen our understanding of the raw-space failure.
> > >
> > > To directly test the reviewer's suggestion, we tried both Gaussian and Laplacian kernels on raw ImageNet pixels, and varied the temperature/bandwidth over a range of values, to vary the degree of tail-heaviness. Empirically, this did not resolve the issue: neither a heavier-tailed kernel nor tuning the bandwidth made raw-space ImageNet training work well.
> > >
> > > We think this is informative because the issue is not identifiability. In the Gaussian-kernel setting, as discussed in our previous response, identifiability holds. Moreover, the same raw-space formulation works on low-dimensional toy data and robotics. This suggests that the main obstacle in high-dimensional raw space is practical estimation rather than population-level identifiability.
> > >
> > > Our current interpretation is that the bottleneck is the **variance of Monte Carlo drift estimation under uniform sampling in high dimension**.
> > >
> > > Heuristically, if nearby points around  $x$ have very small probability under the sampling distribution, then making their contribution non-negligible requires inverse-probability-like reweighting, which can substantially increase estimator variance.
> > >
> > > This explains why changing the kernel alone does not empirically solve the problem. A heavier tail does not change the proposal distribution: the informative nearby points are still too rare under the sampling distribution, so the Monte Carlo estimate of $V$ remains high-variance. We view this as an important limitation of the current Monte Carlo estimation of $V$ in raw space, rather than evidence against the drifting framework itself. Better proposal distributions or alternative drift estimators may be promising future directions, but we leave that to future work.
> > >
> > > By contrast, feature space acts as a practical workaround: semantically similar samples are much more concentrated, so “effective neighbors” are no longer as rare. Consistent with this view, in feature space both Gaussian and Laplacian kernels work on ImageNet, with Laplacian only modestly better in our ablation (8.46 vs. 9.19 FID).
> > >
> > > More broadly, we think this helps contextualize the feature-space results: on ImageNet, once drift estimation is carried out in a representation space with better geometry, the method becomes viable and strong empirically, and Table 3 further suggests that it improves with stronger feature spaces.
> > >
> > > So we agree with the reviewer that, for high-dimensional natural images, the representation space is doing important work in our current ImageNet instantiation. At the same time, our takeaway is more specific: the current difficulty seems to be **raw-space drift estimation in high dimension**, rather than a failure of the drifting formulation itself.

---

### Official Review · Reviewer_yeho · 2026-03-10

**Soundness:** 3
**Presentation:** 3
**Significance:** 4
**Originality:** 4
**Overall Recommendation:** 5
**Confidence:** 5

**Summary:**

The paper proposes drifting models - a novel perspective on training generative models via the training-time evolution of the pushforward distribution. The authors introduce a general framework and propose a particular semi-parametric instantiation that demonstrates strong single-step results on ImageNet in both latent and pixel spaces.

**Compliance With Llm Reviewing Policy:**

Affirmed.

**Final Justification:**

My concerns have been well addressed, and I recommend acceptance. I encourage the authors to clearly present the provided discussions and results in the final revision.

**Key Questions For Authors:**

I am generally positive about the submission and would be willing to see it accepted. However, I would highly appreciate if my two main concerns discussed in the Weaknesses are addressed:

1) I would highly encourage the authors to discuss the connections with existing implicit generative approaches (DMD, GANs, etc) and suggest reframing the claim that the work presents a conceptually novel paradigm rather than a novel unified perspective.

2) I would also kindly ask the authors to eliminate the leakage of ImageNet classes to the feature extractor and report the final results without classification finetuning. I am totally okay if the method does not provide the best FID ever but it is critical to me that the evaluation is fair. Comparisons in terms of FDD (Dino FD) also would be useful.

**Limitations:**

Yes. In Section 6, the authors clearly discuss limitations and future directions. As a recommendation, I would suggest two additional discussions:

1) future work on potential parametric variants of drifting models, and how the drifting perspective could extend DMD-like approaches or vice versa.

2) future work on scaling this perspective (and the specific instantiation proposed in the paper) to text-to-image / video.

**Strengths And Weaknesses:**

---

## Strengths

* The paper introduces a novel and interesting generative modeling formulation that unifies many existing paradigms. I believe the drifting perspective is hightly valuable to the field and may facilitate notable future research

* The proposed framework is clearly presented. The paper provides many useful discussions, supporting illustrations and algorithms.

* The proposed instantiation investigates multiple design choices which are well motivated and carefully ablated.

* To my knowledge, this is the first work demonstrating strong generative results on ImageNet when training a single-step generator from scratch using a semi-parametric approach (feature extractor + non-parametric loss).

* The method shows hightly promising results for both pixel-space and latent-space generation.

---

## Weaknesses

---

### Novelty & Missing connections

I believe the main idea of “the training-time evolution of pushforward distributions” is applicable to all implicit generative models (GANs, DMD, SiD, MMD-based models, etc). Therefore, I would consider reframing drifting models as a unifying perspective of various implicit models rather than as a fundamentally novel paradigm.

For example, the proposed drifting model instantiation can be directly connected to a non-parametric DMD using ideal denoisers [1] (Appendix B3). I believe it can be shown that Eq (11) with the proposed Gaussian kernel is identical to DMD with non-parametric real and fake scores. Interestingly, DMD implements their surrogate loss exactly the same way as in Eq(6). I believe this connection can be quite productive to improve and further generalize drifting models.

GANs also can be treated as parametric drifting models where the discriminator gradient also pushes the generator towards the data distribution. If considering an optimal discriminator in the classical GAN formulation $ D^* = \frac{p_{data}}{p_{data} + p_\theta} $ and substituting it to the generator loss gradient: $\nabla_x \log (1 - D^*(x)) = \nabla_x \log p_{\theta}(x) - \nabla_x \log (p_{data}(x) + p_\theta(x))$ - we also obtain a drifting field $V(p_{data}, p_\theta)$ that equals to 0 when $p_\theta = p_{data}$.

In future work, these connections allow describing drifting models via minimization of various probabilistic measures. Moreover, these connections also allow highlighting the proposed non-parametric instantiation of drifting models in contrast to previous parametric alternatives.

[1] Elucidating the Design Space of Diffusion-Based Generative Models. 2022

---

### Evaluation

I have one important concern that the reported FID results may be largely inflated due to ImageNet classification finetuning (Table 3). This setting is then further scaled to obtain the final FID results in Table 4.

Using ImageNet-tuned feature extractors is a well-known issue [1], as it can result in improved FID without a corresponding improvement in actual fidelity. I also inspected the visual examples in the Appendix, and it is hard to confirm if they indeed correspond to FID=1.54.

[1] The Role of ImageNet Classes in Fréchet Inception Distance, ICLR2023

---

### Presentation (Minor)

Drifting models are introduced via “the evolution of the training-time pushforward” which, in my opinion, is an overly complicated and somewhat confusing way to describe simple generator training or an implicit generative model.

While I respect the authors’ vision in how they present their work, I would recommend connecting their terminology with the common one early in the paper to avoid potential confusion by readers.

In addition, Figure 1 does not seem quite informative to me. Basically, it illustrates that a generator converges during training that is not a big insight.

---

### Classifier-free guidance (Minor)

Equation 16 is somewhat confusing from a theoretical perspective. It seems that $q(\cdot | c)$ can be negative, which is not the case for CFG operating on log-probs. I recommend adding a short discussion clarifying this point and whether it could pose any practical issues.

---

> ### Author Rebuttal · Authors · 2026-03-29
>
> We sincerely thank reviewer yeho for the thoughtful feedback. We are encouraged that the reviewer finds the drifting perspective valuable and the empirical results promising. The suggestions are highly helpful, particularly on (i) positioning relative to prior implicit generative approaches and (ii) fairer evaluation with respect to feature encoders and class information.
>
> ---
>
> ### Q1 / W1. Positioning relative to prior implicit generative models (DMD, GANs, etc.)
>
> We thank the reviewer for this suggestion, which we find important and productive. We agree that a clearer framing is to present drifting as a unified $V$-centric perspective on implicit generative modeling. Our intent is not to deny connections to prior approaches, but to make the drifting field itself explicit as the central design object.
>
> Many implicit generative models can be interpreted as inducing drifting fields:
>
> - **GANs**: $V(x)=\nabla_x \log p_{\text{data}}(x)-\nabla_x \log(p_{\text{data}}(x)+p_\theta(x))$
> - **DMD**: $V(x)=\mathbb E_{t,x_t}[v_{\text{real}}(x_t,t)-v_{\text{gen}}(x_t,t)]$
> - **General divergence** $\min_q L(p,q)$: $V=\nabla_x \frac{\delta L}{\delta q}$
>
> Under this view, our contribution is to elevate $V$ as an explicit designable object, which is more general and enables normalizations, aggregation across features, and modified equilibria. We also show that a non-parametric instantiation works at ImageNet scale without alternative training or using pretrained diffusion models.
>
> **Regarding the DMD connection.** We agree that this connection should be stated explicitly. We plan to discuss it in the revision as follows:
>
> With a Gaussian kernel $k_\sigma(x,y)=\exp(-\|x-y\|^2/2\sigma^2)$, the mean-shift drift field simplifies to
> $$
> V_{p,q}(x)=\sigma^2\left[\nabla_x \log \hat p_\sigma(x)-\nabla_x \log \hat q_\sigma(x)\right].
> $$
> Thus, in this Gaussian-kernel mean-shift special case, the drifting field is a score-difference field of Gaussian-smoothed densities.
>
> Shared with DMD, under Gaussian kernel: a smoothed score-difference updte direction $\nabla \log \tilde p-\nabla \log \tilde q$.
>
> **Framework differences:** our formulation is more general: by making $V$ explicit, it supports aggregation across temperatures and feature spaces under a shared zero-drift condition, normalizations, and modified equilibria.
>
> **Practical differences:** (1) For field, we compute the field on clean samples $x$, often in feature spaces and with non-Gaussian kernels, whereas DMD uses score differences on noisy samples $x_t$, amortized over $t$, directly in raw space. (2) For estimation, our ImageNet instantiation computes $V$ from samples, whereas DMD learns score estimators with alternative training and initialize from pretrained diffusion models.
>
> ---
>
> ### Q2 / W2. ImageNet classification leakage in the feature encoder
>
> We appreciate the reviewer raising this important concern. We fully agree that classification leakage could affect FID, and evaluatiion (1) without classification finetuning (2) on Dino-FD is important.
>
> **Existing experiment in paper.** In Table 3, MAE feature extractors *never* see class labels during training except the last row. The FID without classification finetuning for the B-size model at 100 epochs is 4.28.
>
> **New experiment — L-size model without CLS finetuning.** Under the same never-see-class-labels MAE setting, we train an L-size model:
>
> | Epochs | With CLS ft. | Without CLS ft. |
> |--------|--------------|-----------------|
> | 320    | 1.96         | 2.31            |
> | 640    | 1.68         | 2.02            |
> | 1280   | 1.53         | 1.72            |
>
> The gap narrows with longer training, and the no-CLS run remains competitive.
>
> **DINO-FD (classification-free metric).** For L-size model in paper, we evaluate DINO-FD: **96.5** at CFG 2.0 (and 143.4 at CFG 1.0). Comparing with 1-step models, StyleGAN-XL reports 214.9, DMF L/2 199.7, and DMF XL/2 122.3.
>
> We will include the no-CLS and DINO-FD results in the revision. We appreciate the reviewer’s feedback in prompting these experiments.
>
> ---
>
> ### W3. Presentation and terminology
>
> We agree that the current wording and Figure 1 may overemphasize the generic fact that generators evolve during training. Our intent was to highlight that this evolution is driven by a drifting field $V$, which defines the equilibrium condition and training objective. We plan to revise Figure 1 to make the role of $V$ explicit.
>
> ---
>
> ### W4. CFG equation (Eq. 16)
>
> Our CFG formulation uses linear interpolation on probabilities, not log-probabilities as in diffusion CFG, so $q(\cdot\mid c)$ can in principle be negative. Empirically, we observe stable training and the expected quality-diversity tradeoff. We will clarify this distinction in the revision.
>
> ### Suggested Future Work
>
> We agree. Under the unified view, learning parametric drifting field is a natural extension, and scaling this perspective to text-to-image / video is an exciting direction.

---

> > ### Author Rebuttal · Reviewer_yeho · 2026-04-01
> >
> > I sincirely thank the authors for the detailed rebuttal and appreciate for addressing my concerns. I also carefully read other reviews and author responses and, overall, remain positive about the submission and vote for acceptance. (Increased my score accordingly)
> >
> > Nevertheless, I would like to emphasize that it is important to me that w/ CLS ft + FID results are removed from the main tables while also including DINO-FD there in the revision to mitigate further FID abuse in follow-up works trying to build on the proposed approach.

---

### Official Review · Reviewer_We7j · 2026-03-10

**Soundness:** 2
**Presentation:** 4
**Significance:** 2
**Originality:** 2
**Overall Recommendation:** 4
**Confidence:** 4

**Summary:**

The paper proposes Drifting Models, a generative modeling framework that shifts the distribution transformation process from sampling time to training time. Instead of iteratively converting noise to data during inference (as in diffusion or flow-based models), the method learns a single-step generator by training the model with a drifting field that guides generated samples toward the data distribution. When the model distribution matches the data distribution, the drift vanishes, defining the training objective. As a result, the trained model can generate samples in one forward pass while achieving competitive performance on benchmarks such as ImageNet 256×256.

**Compliance With Llm Reviewing Policy:**

Affirmed.

**Final Justification:**

Thanks for the authors rebuttal. I slightly raised the score

**Key Questions For Authors:**

If the connections mentioned above indeed hold, an important follow-up question is why a non-parametric estimation of the score would be preferable in this setting. The method appears to perform well on the ImageNet benchmark after exploring different feature extractors, as shown in the appendix. However, it remains unclear whether this approach would still hold for larger and more diverse datasets. In other words, it would be helpful to understand whether the method is truly general, beyond the narrative of the so-called drifting model.

Furthermore, I am personally not fully convinced by the motivation illustrated in Figure 1. For most generative models, one can draw a similar figure showing that the generated distribution gradually approaches the target distribution during training. If this type of visualization is used as the defining intuition, it raises the question of whether almost any generative model could be interpreted as a drifting model under such a perspective. Clarifying what fundamentally distinguishes the proposed framework from standard generative training dynamics would strengthen the presentation.

**Limitations:**

see weakness

**Strengths And Weaknesses:**

Strengths:

1. The paper is well-written and easy to understand.

2. The experiments are intensive and comprehensive.

Weakness:

1. Connection to score estimation: The drifting field appears closely related to kernel-based estimators of the score function (e.g., mean-shift–type estimators). In particular, the attraction–repulsion structure resembles estimating the difference between the data score and the model score. Clarifying whether the drifting field can be interpreted as an estimator of the score (or score difference) would help position the method within the broader literature on score-based generative modeling and gradient flows.

2. Similarity to CACR [1] : The loss formulation and the attraction–repulsion structure of the drifting field appear similar to the contrastive objective used in CACR, where softmax/kernel weights over pairwise distances are used to construct weighted positive and negative terms. It would be helpful if the authors could explicitly discuss the similarities and differences between the proposed loss and the CACR objective.

3.	Relation to Distribution Matching Distillation (DMD) [2]: The update direction induced by the drifting field resembles the score-difference update used in DMD, where samples are moved according to the difference between the data score and the model score and the gradient is backpropagated through the generator. Furthermore, the implementation also appears conceptually similar, and the score term used in DMD is itself closely related to the score-estimation perspective discussed in point 1. In other words, the two approaches may mainly differ in how the score (or score difference) is estimated. The authors should clarify the conceptual and practical differences between their approach and DMD, and explain whether the proposed method can be viewed as an alternative estimator of the same underlying update direction.

[1]https://arxiv.org/abs/2105.03746

[2]One-step Diffusion with Distribution Matching Distillation

---

> ### Author Rebuttal · Authors · 2026-03-29
>
> We sincerely thank Reviewer We7j for the thoughtful and insightful feedback. We are glad that the reviewer finds the paper well-written and the experiments comprehensive. The review raises the following main questions: (a) connections to score estimation and DMD, (b) similarity to CACR, (c) Figure 1 motivation, and (d) generality beyond ImageNet. We address each point below.
>
> ---
>
> ### W1 / W3. Connection to score estimation and relation to DMD
>
> We sincerely appreciate the feedback. We agree that this is an important connection, and we plan to include the discussion below:
>
> For our method, in the Gaussian-kernel mean-shift special case,
> $$
> V_{p,q}(x)=\sigma^2\left[\nabla_x \log \hat p_\sigma(x)-\nabla_x \log \hat q_\sigma(x)\right],
> $$
> which provides a score-difference interpretation in that restricted setting and helps explain a similarity in the update direction with DMD.
>
> However, this interpretation does not imply equivalence to score-difference methods such as DMD. In particular, our method defines the drifting field on clean samples $x$ and constructs $V$ directly from samples (often in learned feature spaces), whereas DMD evaluates score differences on noisy samples $x_t$ and involves an expectation over noise levels $t$.
>
> Moreover, the realized field used in practice differs further: our implementation supports non-Gaussian kernels, batch-level sample  normalization, $V$-normalization, and aggregation across multiple feature spaces. These ingredients make the realized update field in practice meaningfully different from a standard score-difference construction.
>
> We plan to revise the paper to clarify both (i) the connection to score-difference directions in a special case, and (ii) the differences in the realized field and training in practice.
>
> ### Q1. Parametric vs Non-parametric estimation
>
> We do not claim that non-parametric estimation is universally preferable. Rather, in our setting it is a particularly natural way to instantiate the drifting field: it constructs sample drifts directly from data/model samples, without alternating score-network training or relying on a pretrained diffusion teacher.
>
> It is also especially convenient for our formulation because it lets us define and aggregate drifting fields directly across multiple feature spaces; a parametric score-based route would require learning separate score estimators for each representation space, which is substantially heavier.
>
> Empirically, Table 3 suggests a scaling-like trend: stronger feature spaces lead to better performance. At the same time, our current evidence suggests that for high-dimensional natural images this route still depends heavily on informative feature geometry, and achieving equally strong performance in raw space remains an open challenge. We will clarify this scope and tradeoff in the revision.
>
> ---
>
> ### W2. Similarity to CACR
>
> Thank you for pointing out this connection. We agree that it is important and should be discussed more directly.
>
> The attraction–repulsion structure is indeed related. In particular, CACR also uses weighted positive and negative interactions estimated non-parametrically, with separate normalizations. A key difference is the role this mechanism plays. In CACR and related contrastive representation-learning methods, attraction–repulsion is used to learn strong embeddings. In our formulation, the induced field is treated as an explicit update direction for generated samples, and the zero-drift condition is used as the generative matching principle.
>
> So we agree that the mathematical structure is related, but our use of it is different: not as a representation-learning objective, but as a sample-update field for training a generator toward distributional equilibrium. We will add this discussion explicitly in the revision.
>
> ---
>
> ### Q1 / Q2. Generality and Figure 1 motivation
> We appreciate the reviewer’s question about generality. Our current evidence supports a narrower claim than a broad “universal” one: the drifting formulation itself is not tied to ImageNet, since it already works without pretrained features in toy problems and robotics, where raw-space distances are informative. The ImageNet result is one particular high-dimensional instantiation, and in that regime our current evidence suggests that informative feature spaces are important. We agree that demonstrating broader generality on larger and more diverse datasets is an important direction for future work, and we plan to state the current scope more clearly in the revision.
>
> We also appreciate the feedback on Figure 1. The current wording and figure might overemphasize the generic fact that generators evolve during training. What we intended to highlight instead is that our formulation explicitly parameterizes this training-time evolution by a drifting field $V$, which is then used to define equilibrium conditions and the training objective. We plan to revise Figure 1 to make this role of $V$ more explicit.

---

> > ### Author Rebuttal · Reviewer_We7j · 2026-04-01
> >
> > Thanks for the rebuttal. Please do include previous discussion in the final revision.
> >
> > Also, I just came across a recent paper related to our discussion [1]. I understand that it was released after your submission, and I am not bringing it up to affect the evaluation of the current paper. Rather, I mean it only as a gentle suggestion: a brief discussion of its relation to your work could be helpful for a broader audience. For clarity, I am not an author of that paper.
> >
> > I also experimented with my own implementation of the drift model, and my impression is that it is quite sensitive to the dataset scale, data dimension, and feature extractor. Because of this, I think it would be important to add a few more sentences about these practical sensitivities in the main paper, or, it would be better for the authors to provide the rule of thumb for triaining drift model.
> >
> > [1] https://arxiv.org/abs/2603.07514

---

> > > ### Author Response · Authors · 2026-04-01
> > >
> > > We thank the reviewer for pointing out the related work. We will incorporate the relevant discussion from the rebuttal into the final revision.
> > >
> > > We also thank the reviewer for the helpful follow-up comments, which we respond to below.
> > >
> > > ---
> > >
> > > ### Q1. Recent work
> > >
> > > We also appreciate the pointer to the recent related work [1]. Since it appeared after our submission, we did not discuss it in the current draft. We agree that a brief discussion of its relation to our work would be helpful for readers, and we will include it in the final version. At a high level, we view it as complementary to our paper: our work focuses on the drifting formulation and a practical large-scale non-parametric instantiation, while this follow-up work further studies the connection between drifting and score-based perspectives.
> > >
> > > ---
> > >
> > > ### Q2. Rule of thumb
> > >
> > > We also thank the reviewer for experimenting with the method and for raising the question of practical guidance. Our experience is that the method is reasonably robust across the settings we explored, although a few normalization and stabilization choices are important in practice. We agree that these details are not stated clearly enough in the current draft, and that adding a short practical discussion / rule-of-thumb paragraph would improve clarity and reproducibility.
> > >
> > > In particular, we use the following practices:
> > > 1. **Feature and drift normalization.** For each feature, we normalize the per-coordinate scale to be on the order of $\Theta(1)$, and similarly normalize the per-coordinate scale of the drifting field $V$ to $\Theta(1)$. We then apply an MSE loss on $x - \mathrm{sg}(x+V)$, so that each loss term is also on the order of $\Theta(1)$. We also rescale the temperature by the average inter-point distance, so that temperature choice can be shared across different features. More details are provided in Appendix A.6.
> > >
> > > 2. **Multiple temperatures and features.** After the normalization above, using multiple temperatures and multiple feature layers has generally been helpful in our experience. In our experiments, we typically use a temperature mixture $[0.02, 0.05, 0.2]$, together with features from multiple layers (e.g. every 2 layers) of the encoder. Since the feature encoder is only run  once, this adds little computational overhead.
> > >
> > > 3. **Early-stage stabilization.** With AdaLN-zero initialization of DiT's last layer, the generator can stay near low-magnitude outputs early in training. We found it helpful to include an additional norm feature, e.g. the average magnitude of $x^2$ across channels, which helps the model leave this regime more quickly.
> > >
> > > 4. **Learning-rate selection.** In practice, we choose the learning rate based on the pre-normalization magnitude of $V$, which we have found to be a useful proxy for the loss.
> > >
> > > We will also clarify two implementation details that are easy to overlook:
> > > (i) the feature encoder should receive the input normalization it was designed for; for example, if an encoder expects inputs in the range $[-1,1]$, feeding inputs in $[0,1]$ can lead to a substantial mismatch; and
> > > (ii) when features are scale-invariant (e.g. when GroupNorm is applied on the input), a global loss on raw data (which we include in our experiments), or an auxiliary norm feature as above, is important.
> > >
> > >
> > > Most of the choices above are simple and well motivated by two principles: first, it is important to keep different loss terms on comparable scales; second, once scales are matched, combining multiple informative losses generally helps. We agree that these points should be stated more explicitly in the paper. In our experience, with the practices above in place, training is generally stable and requires limited tuning, typically only the learning rate. We appreciate the reviewer for highlighting that such practical guidance would be useful, and we will include it in the final paper.
> > >
> > > [1] https://arxiv.org/abs/2603.07514

---

### Official Review · Reviewer_XgXL · 2026-03-11

**Soundness:** 3
**Presentation:** 4
**Significance:** 2
**Originality:** 3
**Overall Recommendation:** 4
**Confidence:** 4

**Summary:**

This paper introduces Drifting Models, a generative modeling framework that shifts the iterative pushforward process from inference time to training time. The central mechanism is a drifting field $V$ that governs how generated samples move during training. This field is designed to be anti-symmetric ($V_{p,q} = -V_{q,p}$), which guarantees $V=0$ when the generated distribution $q$ matches the data distribution $p$. The method uses a kernel-based attraction-repulsion formulation: generated samples are pulled toward real data (positives) and pushed away from other generated samples (negatives). The training objective minimizes $\|V\|^2$ via a stop-gradient fixed-point iteration, and the approach naturally yields one-step (1-NFE) inference. On ImageNet $256 \times 256$, the method reports 1.54 FID in latent space and 1.73 FID in pixel space, outperforming prior single-step generators.

**Compliance With Llm Reviewing Policy:**

Affirmed.

**Final Justification:**

The authors have adequately addressed most of my concerns in their rebuttal. I maintain my score.

**Key Questions For Authors:**

1. Can you train with CFG completely removed ($N_{\text{unc}}=0$, $\alpha$ fixed at 1) and report FID? Or run an unconditional experiment (say CIFAR-10)? This is the most direct way to isolate the drifting contribution.

2. Have you tracked $\|V_{\text{drifting}}\|$ vs $\|V_{\text{cfg}}\|$ over the course of training? If $V_{\text{cfg}}$ dominates late in training, that would be consistent with the Decoupled DMD [R3] and Self-Evaluation [R2] finding that distribution matching is a regularizer while CFG drives quality.

3. How much total compute (GPU hours) do the final models take, including MAE pre-training?

4. A discussion mentioned in the above weaknesses' references would help position what's new in the drifting formulation vs. what's shared with these methods.

**Limitations:**

The authors acknowledge the converse-direction gap and the feature encoder dependence.

**Strengths And Weaknesses:**

Strengths:

- The idea of moving the iterative process from inference to training is interesting. The drifting field formulation is easy to understand and authors provide several propositions to understand its properties.

- The one-step results are appealing: 1.54 FID with 1 NFE in latent space, 1.73 in pixel space. The pixel-space result is especially impressive given the FLOPs.

- Here are several useful ablation studies. The anti-symmetry destructive study (Table 1) is convincing, proving this property is essential for constructing an appropriate drifting field. The positive/negative allocation study (Table 2) and feature encoder comparison (Table 3) are also useful to understand when the proposed method can work.

Weaknesses:

- My main concern is that the paper never disentangles the drifting field from the CFG signal, and I believe the CFG signal may be doing most of the heavy lifting in later training.

The negative samples have two sources: generated data $x_{\text{gen}} \sim q_\theta(\cdot|c)$ and unconditional real data $y_{\text{unc}} \sim p_{\text{data}}(\cdot|\varnothing)$. So $V$ splits into two terms: $V(x) = V_{\text{drifting}}(x) + V_{\text{cfg}}(x)$. Both terms have the same kernel-weighted form, computing $\tilde{k}(x,y^+) \tilde{k}(x,y^-)(y^+ - y^-)$ averaged over positive-negative pairs, where $y^+$ are real class-conditional samples. For $V_{\text{drifting}}$, the negatives $y^-$ are generated samples from $q_c$, weighted by $(1-\gamma)$. For $V_{\text{cfg}}$, the negatives are unconditional real data from $p_\varnothing$, weighted by $\gamma$.


$V_{\text{drifting}}$ is anti-symmetric and goes to zero when $q = p$. $V_{\text{cfg}}$ is a kernel-weighted cond-uncond signal. It does NOT go to zero when $q = p$ because $p_{\text{data}}(\cdot|c) \neq p_{\text{data}}(\cdot|\varnothing)$.

The loss is $\|V\|^2$ with stop-gradient, so the gradient on the generator is $\partial L / \partial \theta = -2V^\top \cdot \partial f_\theta / \partial \theta$. As $q \to p$ during training, $V_{\text{drifting}} \to 0$ and $V_{\text{cfg}}$ stays. Late in training, the gradient is mostly $V_{\text{cfg}}$.

This is not a speculative worry. Decoupled DMD [R3] did exactly this decomposition for DMD training and showed the CFG component drives quality while distribution matching is just a regularizer. Self-Evaluation [R2] takes it further: it shows that a single denoising step producing a cond-uncond correction is enough to train one-step generators, with no explicit distribution matching at all. The parallel here is hard to miss: $V_{\text{drifting}}$ is the distribution matching (regularizer), $V_{\text{cfg}}$ is the CFG signal (driver).

Some numbers: at $\alpha = 2$ with $N_{\text{neg}}=64, N_{\text{unc}}=16$, the unconditional weight is $w \approx 3.94$, and the cond-uncond term accounts for roughly 50% of the negative signal. At $\alpha = 4$, it's around 75%.

Every configuration in Table 8 includes CFG. The paper never trains with $N_{\text{unc}}=0$ and $\alpha$ fixed at 1. The claim that best FID occurs at $\alpha=1.0$ only refers to inference time; during training, the $\alpha > 1$ iterations keep feeding $V_{\text{cfg}}$ gradients into the shared parameters. Without a CFG-free training ablation, the contribution of the drifting formulation itself is unclear.

- The method does not work without a pretrained feature encoder. The authors say so explicitly (page 7). Looking at Table 3, the encoder quality pretty much determines the FID: widening the MAE and adding classification fine-tuning takes FID from 8.46 to 3.36. The kernel $k(x,y) = \exp(-\|x-y\|/\tau)$ needs samples to be meaningfully close or far in the feature space. In raw pixel/latent space, high-dimensional distances concentrate and the kernel becomes almost uniform. So the feature encoder isn't a nice-to-have; it's what makes the whole approach viable.

- The theory has a gap that matters. Anti-symmetry gives $p = q \Rightarrow V = 0$, which is trivial. The other direction ($V \approx 0 \Rightarrow p \approx q$) gets only a heuristic argument in Appendix C.

- There are a lot of engineering components: feature normalization, drift normalization, three temperatures, CFG-conditioning, a custom MAE with classification fine-tuning. Table 1 shows anti-symmetry is necessary, but that doesn't tell us whether the drifting formulation is sufficient or whether the performance comes from the full engineering stack.

- The related work discussion misses important connections.

Any distribution matching method (DMD [R4], Diff-Instruct [R5]) can be seen as computing a per-sample "drift" via the gradient of a divergence. The paper only mentions these as "distillation-based methods" without engaging with the mechanistic similarity.

A better example of "bringing test-time iteration into training" is methods that denoise the model's own outputs and regress toward the denoised version. Both Terminal Velocity Matching [R1] and Self-Evaluation [R2] did this: take a generated sample, apply one denoising step (using the model's own score), and use the result as a stop-gradient target. That's the same $\text{sg}(x + \Delta)$ structure as this paper. Both methods turn training-time iterative refinement of the pushforward distribution into a single-step generator.

- [R1] Zhou etal. "Terminal Velocity Matching." arXiv:2511.19797
- [R2] Yu etal. "Self-Evaluation Unlocks Any-Step Text-to-Image Generation." arXiv:2512.22374
- [R3] Liu etal. "Decoupled DMD: CFG Augmentation as the Spear, Distribution Matching as the Shield." arXiv:2511.22677
- [R4] Yin etal. "One-step Diffusion with Distribution Matching Distillation." arXiv:2311.18828.
- [R5] Luo etal. "Diff-Instruct: A Universal Approach for Transferring Knowledge From Pre-trained Diffusion Models." arXiv:2305.18455.

---

> ### Author Rebuttal · Authors · 2026-03-29
>
> We sincerely thank Reviewer XgXL for the technically engaged feedback. We are glad that the reviewer finds the idea of moving the iterative process from inference to training interesting, considers the one-step results appealing, and finds the ablations useful. The main concerns are helpful, particularly (1) the role of CFG, (2) related work connections, and (3) the theory gap. We address each point below.
>
> ---
>
> ### W1 / Q1 / Q2. The effect of CFG
>
> We agree that disentangling CFG is important. The experiments below isolate the role of drifting.
>
> **No-CFG training ablation.** We trained the model with entirely remaining CFG ($N_{\text{unc}}=0$, $\alpha \equiv 1$, B/2,100 epochs):
>
> | Setting | With CFG | No CFG |
> |---------|----------|--------|
> | Baseline (MAE-256) | 8.46 | 15.99 |
> | Stronger features (MAE-640) | 3.36 | 4.04 |
>
> This ablation suggests that the drifting objective remains meaningful on its own, while CFG provides additional gains and is more important in weaker-feature settings.
>
> **Norm tracking.** We tracked $\lVert \hat V_{\text{drifting}} \rVert / \lVert \hat V_{\text{cfg}} \rVert$ under setting ($\alpha=2$, $N_{\text{unc}}=16$). The ratio converges to approximately **0.75**, suggesting that $V_{\text{drifting}}$ remains substantial.
>
>
> **Why does $V_{\text{drifting}}$ not vanish?** We found Eq. (15) as written not clear enough. We wrote
> $$
> q(\cdot \mid c)\triangleq (1-\gamma)q_\theta(\cdot \mid c)+\gamma p_{\text{data}}(\cdot \mid \varnothing).
> $$
> In implementation, however, the generator distribution is indexed by the CFG scale, i.e. $q_\theta(\cdot \mid c,\alpha)$. For a given $\alpha$, the generated distribution $q_\theta(\cdot \mid c,\alpha)$ is attracted by $p_{\text{data}}(\cdot \mid c)$ and repelled by
> $$
> (1-\gamma)q_\theta(\cdot \mid c,\alpha)+\gamma p_{\text{data}}(\cdot \mid \varnothing).
> $$
> The corresponding equilibrium is
> $$
> q_\theta^*(\cdot \mid c,\alpha)=\alpha p_{\text{data}}(\cdot \mid c)-(\alpha-1) p_{\text{data}}(\cdot \mid \varnothing)\neq p_{\text{data}}(\cdot \mid c).
> $$
> Thus the drifting term does not vanish on its own; rather, the two terms balance in the total field. We will revise the paper to clearly distinguish $q_\theta(\cdot \mid c)$ from $q_\theta(\cdot \mid c,\alpha)$. We sincerely thank the reviewer for noticing this ambiguity.
>
> ---
>
> ### W5 / Q4. Missing related work connections
>
> Thanks for pointing out these important connections. We agree that these works are related. We plan to add following citations and discussions:
>
> DMD, Diff-Instruct, and GANs can be viewed as inducing different per-sample drifts (see details in response to Reviewer yeho). A key difference is that our formulation treats the drifting field $V$ explicitly, which supports normalizations, aggregation across feature spaces, and modified equilibrium. Our ImageNet instantiation is also non-parametric and does not use alternating training or a pretrained diffusion model.
>
> TVM and Self-E share a related $\operatorname{sg}(x+\Delta)$ structure, and enable long jumps along the FM trajectory. TVM builds $\Delta$ by perturbing FM endpoint, while Self-E builds $\Delta$ using implicit classification scores. In our method, by contrast, $\Delta$ comes from an explicit drifting field and does not rely on an FM trajectory.
>
> ---
>
> ### W3. Theory gap ($V=0 \Rightarrow p=q$)
>
> We agree this is important. Inspired by this question and Reviewer dwNu's suggestion, we developed two identifiability results during rebuttal:
>
> 1. **Gaussian kernel (general):** With a Gaussian kernel $k$, the mean-shift drifting field equals
>    $$
>    \sigma^2\bigl[\nabla_x \log \hat p_\sigma(x)-\nabla_x \log \hat q_\sigma(x)\bigr].
>    $$
>    If $V=0$ everywhere, then $\hat p_\sigma=\hat q_\sigma$, and since Gaussian convolution is injective, $p=q$.
>
> 2. **Laplacian kernel, Gaussian targets:** For our practical kernel $k(x,y)=\exp(-\lVert x-y\rVert/\tau)$ with $P,Q$ both Gaussian, we derive
>    $$
>    V_{p,q}(ru)\to (\mu_p-\mu_q)+(\Sigma_p-\Sigma_q)u/\tau
>    $$
>    as $r\to\infty$, which determines both the mean and covariance for Gaussian family.
>
> We plan to include these in the revised paper. The general converse for arbitrary fields remains open.
>
> ---
>
> ### W6 / Q3. Compute
>
> We trained on TPU v6e, with the following compute:
>
> - *Ablation:* MAE 2.5h + Generator 2.5h on 64 TPU v6e.
> - *Final L-size:* MAE 22h + Generator 37h on 128 TPU v6e.
>
> ---
>
> ### W4. Heavy engineering stack
>
> We appreciate this feedback. We will clarify the organizing principles in the revision: temperatures are combined without sweeping because $V$-normalization puts them on similar scale, shared across experiments; feature/drift normalization aims to make everything unit scale; and MAE variants are ablated in Table 3.
>
> ### W2. Requirement of features
>
> We agree that features are important for out current ImageNet instantiation. Toy and robotics settings work without features, but making ImageNet work without features remains open.

---

> > ### Author Rebuttal · Reviewer_XgXL · 2026-04-04
> >
> > The authors have adequately addressed most of my concerns in their rebuttal. I maintain my score.

---

### Decision · Program_Chairs · 2026-04-30

**Decision:**

Reject

**Comment:**

This paper proposes a generative modelling framework in which the outputs of a one-step generator are pushed to move in the direction of the current generated distribution evolved by a "drifting field", where the latter is defined in such a way that (under asymptotic conditions) the target distribution is a unique stationary point. This is distinct from typical formulations in which a divergence between the generated and target distributions is minimised directly: instead, the target for infinitesimal evolution of the generated distribution is defined explicitly. If the drifting field is the Wasserstein gradient of a divergence, the modelled distribution follows the divergence's gradient flow over the course of training.

The idea to use an MMD-based flow for learning generative models is old (for example, discussion in [Genevay et al., "Learning Generative Models with Sinkhorn Divergences", AISTATS'18] or MMD-GAN), but the use of such ideas in combination with the drifting field method is new and surprisingly effective, as shown in this paper's experiments. The paper also provides some theoretical analysis showing convergence under certain assumptions. The reviewers are mostly positive about the paper, especially the strong one-step generation results on a variety of problems, and about the presentation and clarity of the paper. (I also enjoyed reading the paper and share the positive assessment.)

However, multiple reviewers pointed out the following important weaknesses of the paper:
- Reliance on the feature encoder (could that be learnt jointly, and the method made adversarial?). More experiments, possibly in low-dimensional settings, are encouraged to stress-test the reliance on the encoder.
- Missed related work and placement in context. The addition of analogies and connections, not only contrasts, between the proposed method and existing work -- see the discussion in rebuttal -- would not weaken the paper's contribution, but rather increase its appeal and impact.

The above questions open some interesting directions, but the degree to which these questions begin to be explored in this submission is insufficient. The paper is near the borderline and I strongly encourage a resubmission with improvements along the axes described above.